# HDCS: Hierarchy Discovery and Critic Shaping for Reinforcement Learning with Automaton Specification

**Duo Xu**                                                                 *dxu301@gatech.edu*
*School of Electrical and Computer Engineering*
*Georgia Institute of Technology*

**Faramarz Fekri**                                              *faramarz.fekri@ece.gatech.edu*
*School of Electrical and Computer Engineering*
*Georgia Institute of Technology*

Reviewed    on    OpenReview:    *https: // openreview. net/ forum? id=BGoRme2MfG&referrer= %5BAuthor% 20Console% 5D*

## Abstract

Training reinforcement learning (RL) agents by scalar reward signals is often infeasible when an environment has sparse and non-Markovian rewards. Deterministic finite-state automaton (DFA) provides a streamlined method for specifying tasks in reinforcement learning (RL) that surpass the limitations of traditional discounted return formulations. However, existing RL algorithms designed to address DFA tasks face unresolved challenges, hindering their practical application. One key issue is that subgoals in the DFA may exhibit hidden hierarchical structures, with some macro-subgoals comprising multiple micro-subgoals in certain orders. Without understanding this hierarchy, RL algorithms may struggle to efficiently solve tasks involving such macro-subgoals. Additionally, the sparse reward problem remains inadequately addressed. Previous approaches, such as potential-based reward shaping, often encounter inefficiencies or result in suboptimal solutions. To address these challenges, we propose a novel RL framework designed to uncover the hierarchical structure of subgoals and accelerating the solving of DFA tasks without changing the original optimal policies, short as HDCS. The framework operates in two phases: first, a hierarchical RL method is used to identify the prerequisites of subgoals and build the hierarchy; second, given any task specification (DFA), the subgoal hierarchy is incorporated into task DFA to make a product DFA for the agent to satisfy, and then a simple and novel critic shaping approach is proposed to accelerate the satisfaction of the product DFA without changing optimal policies of the original problem. In addition, the convergence of the critic shaping method is theoretically proved. The effectiveness of HDCS is demonstrated through experiments conducted across various domains. Especially, compared with representative baselines, the critic shaping can have 2X or 3X acceleration on task solving.

## 1 Introduction

In the classical reinforcement learning (RL) framework, the primary objective is to develop a strategy that maximizes a reward function within an unfamiliar environment. Typically, practitioners design a scalar reward function to guide the agent toward desirable behaviors during the learning process. However, scalar rewards often fail to effectively capture the complexity of real-world task specifications (Toromanoff et al., 2019). While some researchers have posited the hypothesis that "the reward function is enough" (Sutton, 2018; Silver et al., 2021), certain tasks remain inherently unrepresentable by scalar rewards alone (Abel et al., 2021).

This work is supported by ARO grant under Award #W911NF-23-1-0146 and a gift by Open Philanthropy.

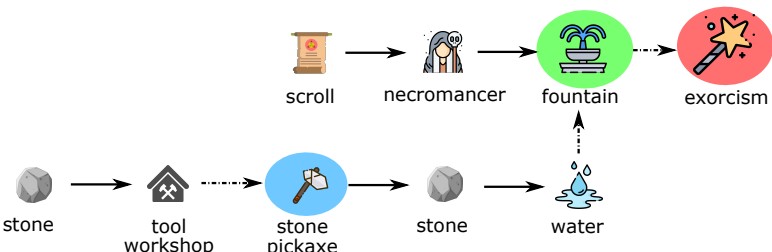

Figure 1: Example of subgoal hierarchy. Every icon represents a subgoal or final goal. The final goal is to get exorcism. The prerequisite of that goal is to first get a scroll, second visit a necromancer, and then build a fountain. Here, building a fountain has its own prerequisites under ordering constraint, including making a stone pickaxe, collecting a stone and going to a place with water. In addition, the prerequisites of making a stone pickaxe consist of first collecting a stone and then visiting a tool workshop. In addition to the fact that stone pickaxe, fountain and exorcism are in level 1 (blue), 2 (green) and 3 (red), all the other subgoals are in level 0 and do not have any prerequisites.

To address these limitations, recent research has increasingly turned to deterministic finite-state automaton (DFA), which subsumes many previous attempts to define specifications that cannot be reduced to scalar Markovian rewards (Abel et al., 2021). These works include Linear Temporal Logic (LTL) (Camacho et al., 2019; Fu & Topcu, 2014; Hasanbeig et al., 2018; Sadigh et al., 2014; Wang et al., 2020; Vaezipoor et al., 2021; Alur et al., 2022; De Giacomo et al., 2020; Voloshin et al., 2023), SPECTRL (Jothimurugan et al., 2019; 2021) and reward machine (Icarte et al., 2022; Corazza et al., 2022; Neider et al., 2021). DFA consists of temporally extended subgoals and can express desired characteristics of future behavior of a system, allowing for precise and flexible task specification (Fersman et al., 2007). Tasks specified using DFA are prevalent in various real-world applications, including robotics, software verification (Bauer et al., 2011), and system control (Tabuada & Pappas, 2003). However, previous works of RL for solving DFA tasks still have two issues remaining unsolved, which could prevent RL algorithms from being deployed into real-world applications.

First, the hierarchical structures of subgoals have been largely overlooked in previous research. In many real-world applications, completing macro subgoals often comprise achieving micro subgoals under ordering constraints. This can form hierarchies of subgoals, where achieving lower-level (micro) subgoals acts as a prerequisite for achieving higher-level (macro) goals. For instance, in the NetHack game (Küttler et al., 2020), which can be regarded as a variant of Minecraft game (Duncan, 2011), subgoals exhibit a layered structure with ordered dependencies. For the example shown in Figure 1, subgoals can be categorized into 4 levels and the prerequisites of every subgoal are under temporal ordering constraints. Before completing the desired goal, the agent has to achieve every subgoal in the prerequisite following the right order. This subgoal hierarchy describing ordering dependencies is typically unknown to the agent at the outset. Without uncovering and leveraging this hierarchy, RL algorithms may struggle to achieve satisfactory learning efficiency.

Second, the issue of sparse rewards in the context of DFA tasks remains insufficiently addressed. Potential-based reward shaping (PBRS) has been employed in previous studies to densify the reward signal (Ng et al., 1999; Grześ, 2017; Jiang et al., 2021; Devidze et al., 2022; Icarte et al., 2022), wherein auxiliary rewards derived from a potential function are added onto the original reward function. However, these reward shaping methods still exhibit certain drawbacks. First, the presence of temporal dependencies in DFA tasks can render the computation of the potential function both difficult and cumbersome, particularly when subgoal prerequisites are incorporated. Second, because DFA tasks typically encompass multiple subgoals over extended temporal stages, incorporating subgoal-specific auxiliary rewards risks diverting the agent's attention to other redundant subgoals, thus destabilizing the agent's learning. Moreover, given that the policy learning is guided by the critic function, the auxiliary rewards must first be assimilated by the critic to subsequently facilitate policy learning—a process that can be inefficient, especially in tasks with lengthy time horizons. Some studies have attempted to decompose DFA tasks into subtasks and address them via hierarchical reinforcement learning (HRL) (Araki et al., 2021; Camacho et al., 2019; Icarte et al., 2022). However, in many practical settings, independently solving each subtask may yield suboptimal solutions.

In this work, we address the aforementioned challenges by introducing a novel framework, referred to as **HDCS**, that discovers hierarchical subgoal structures and expedites DFA task resolution without modifying the

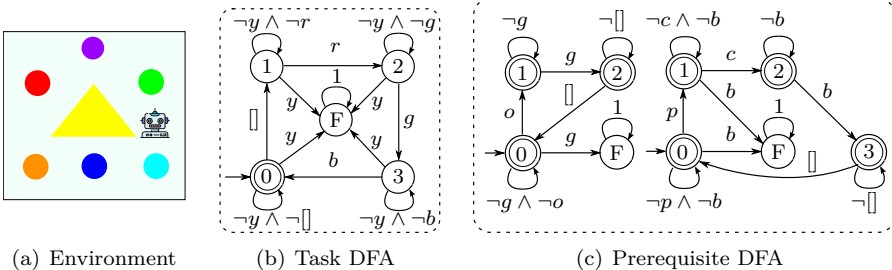

(a) Environment         (b) Task DFA            (c) Prerequisite DFA

Figure 2: (a): the map of environment. (b): the DFA of LTL task formula $\Box(\Diamond(r \wedge \Diamond(g \wedge \Diamond(b)))) \wedge \Box(\neg y)$. (c): The DFAs representing the prerequisites of visiting green ("g") zone and blue ("b") zone. States with double circles are accepting states. State "F" indicates the sink state which can make the task fail. "[]" denotes the empty output of the labeling function. In (c), the left DFA shows that the prerequisite of "g" is to first visit "o", and the right DFA shows that the prerequisite of "b" is to fist visit "p" and then "c". In this work, the task formula is known to the agent while the prerequisites of subgoals are unknown initially.

original optimal policy. The framework proceeds in two stages. First, we propose a hierarchical reinforcement learning (HRL) approach to learn subgoal prerequisites and construct a corresponding subgoal hierarchy. These learned prerequisites are then integrated into the task specification (DFA) through the construction of a product automaton. Second, unlike existing reward shaping methods, we propose shaping the critic function directly to enhance learning efficiency and mitigate the sparse reward problem.

More specifically, we design an auxiliary reward function based on task progress, measured by a reduction in the distance to an accepting state within the task automaton. Whenever the agent progresses to a new stage—i.e., reduces the minimal distance to an accepting state the agent has ever achieved by one hop—the corresponding auxiliary reward is integrated into the critic function via a maximum operator. We term this mechanism *critic shaping*, since it fixes and stabilizes critic values of the target states in all stages. Because this approach bypasses the reward function, it substantially accelerates policy learning, thus alleviating the sparse reward issue in temporal logic tasks. The convergence of the critic shaping method is theoretically proved in Appendix D.

In experiments, we comprehensively evaluate the HDCS framework in terms of subgoal hierarchy discovery and acceleration in solving DFA tasks. The experiments are conducted in 2D-Minecraft, AssemblyZone and Button domains, covering both discrete and continuous action spaces. DFA tasks without and with subgoal hierarchy are both evaluated. The results show that HDCS can learn subgoal hierarchy accurately and expedite the efficiency of solving the DFA task for 2 or 3 times, compared to the state-of-the-art baselines.

## 2 Problem Setting

### 2.1 Definitions

First, we present some basic definitions. An atomic proposition is a variable that takes on a Boolean truth value. The alphabet of symbols is defined as a subset of all possible combinations over a finite set of atomic propositions (denoted as $\mathcal{P}$), i.e., $2^{\mathcal{P}}$. In every working environment, we assume a set of subgoals $\mathcal{G}$ which is a subset of the alphabet, i.e., $\mathcal{G} \subset 2^{\mathcal{P}}$. The subgoals in $\mathcal{G}$ have hierarchical structures which are unknown to the agent initially.

**Deterministic Finite-state Automata (DFA).** In this work, we consider solving tasks specified by DFAs. Specifically, a DFA is defined as a tuple $\mathbb{B} = (\mathcal{B}, \Sigma, T^{\mathbb{B}}, \mathcal{B}^*, a_0)$, where $\mathcal{B}$ is the set of automaton states, the alphabet $\Sigma \subset 2^{\mathcal{P}}$ denotes the alphabet of propositions, driving deterministic transitions in the automaton, the function $T^{\mathbb{B}} : \mathcal{B} \times \Sigma \to \mathcal{B}$ denotes the deterministic transitions of automaton states, $\mathcal{B}^*$ is the set of accepting states, and $a_0$ is the initial state. A *path* $\xi = (b_0, b_1, \ldots)$ is a sequence of automaton states in $\mathcal{B}$ reached through sequential transitions governed by $T^{\mathbb{B}}$. In addition, we assume that DFAs for task specification are only composed by subgoals in $\mathcal{G}$, i.e., $\Sigma = \mathcal{G}$.

**Remark.** We focus on DFAs rather than LTL for two main reasons. First, with finite traces, LTL is strictly less expressive than DFAs—an example being that LTL cannot capture tasks such as "the light switch is

toggled an odd number of times." Second, both DFAs and LTL tasks form countably infinite sets, so any distribution over LTL inherently favors certain subclasses. By contrast, the syntactic structure of DFAs allows these subclasses to be easily distinguished.

We show an example of DFA task in Figure 2. A robot is navigating in the environment with multiple colored zones, as shown in Figure 2(a), where subgoals in $\mathcal{G} = \{r, g, b, o, p, c, y\}$ correspond to whether the agent steps into the red, green, blue, orange, purple, or cyan zones. As shown in Figure 2(b), the task DFA specification in this example asks the agent to oscillate among the red, yellow, and green zones with a fixed temporal order while always avoiding the yellow zone.

**Definition 3.1** (Satisfaction) A path $\xi = (b_0, b_1, \ldots)$ satisfies a DFA $\mathbb{B}$ if an accepting state of $\mathbb{B}$ is visited infinitely often by $\xi$.

**Labeled MDP.** In this work, the environment where the agent is working is formulated as a labeled Markov Decision Process (MDP) $\mathcal{M} = (\mathcal{S}, \mathcal{A}, T^{\mathcal{M}}, \Delta(s_0), \gamma, r, L^{\mathcal{M}})$, with the state space $\mathcal{S}$, the action space $\mathcal{A}$, the state transition $T^{\mathcal{M}} : \mathcal{S} \times \mathcal{A} \to \Delta(\mathcal{S})$ mapping from input state-action pair into a distribution of next states, an distribution of initial states $\Delta(s_0)$, a discount factor $\gamma \in (0, 1)$, a reward function $r : \mathcal{S} \times \mathcal{A} \to [R_{\min}, R_{\max}]$, and a labeling function $L^{\mathcal{M}} : \mathcal{S} \to \Sigma$. The labeling function produces atomic propositions in $\mathcal{P}$ which are true in the input state. If no propositions are true, it produces an empty []. Note that the transition function $T^{\mathcal{M}}$ is *unknown* to the agent. In this work, we assume that the agent is working in *a goal-oriented MDP*, where the labeling function produces propositions which represent subgoals, i.e., $\Sigma = \mathcal{G}$. We assume that the set $\mathcal{G}$ is known to the agent, but the hierarchy of subgoals in $\mathcal{G}$ is unknown initially. The environments used in this work are *deterministic* and the output of $T^{\mathcal{M}}$ is a distribution whose the support is on a single point only.

## 2.2 Subgoal Prerequisites and Hierarchy

**Hierarchy Formulation.** In this work, we assume that subgoals follow a layered structure, where the prerequisites of visiting a subgoal only depend on lower-level subgoals, under some ordering constraints. Specifically, subgoals in level 0, denoted as $\mathcal{G}_0$, can be achieved without any prerequisites. In level $k$, subgoals (denoted as $\mathcal{G}_k$) have prerequisites comprising subgoals from levels 0 through $k-1$. We further assume that subgoals in the level 0 are directly visible in the environment, so that the set $\mathcal{G}_0$ is known to the agent initially. Note that subgoal prerequisites cannot contain higher-level subgoals. Otherwise, circular dependencies can be created, making the involved subgoals unreachable. For example, if the prerequisite of subgoal "a" depends on "b," and prerequisite of "b" includes "a," neither "a" or "b" can be achieved.

This kind of subgoal hierarchy exists in many real-world problems. For instance, Figure 1 illustrates a simplified example from the NetHack game, which is a variant of MineCraft. The goal in the highest level (level 3) is to get "exorcism", and its prerequisites are to first get a scroll, second visit the necromancer, then build a fountain. The subgoal of building a fountain is a level-2 subgoal whose prerequisites are to first make a stone pickaxe, then get a stone and then go to a place with water. In addition, the subgoal of "stone pickaxe" is in level 1 which requires the agent first get a stone and visit the tool workshop. So, every subgoal in level 1, 2 and 3 has prerequisites under temporal ordering constraints. Except "exorcism", "fountain" and "stone pickaxe", all the other subgoals are in level 0.

However, except level-0 subgoals, this hierarchical structure of subgoals is often unknown prior to interacting with the environment. To address this, before tackling the DFA task, we propose to use HRL to first learn subgoal prerequisites to build the subgoal hierarchy. Subsequently, we need to incorporate the learned subgoal prerequisites into the DFA task for the agent to follow. Hence, we represent subgoal prerequisites in the form of DFAs, referred to *prerequisite DFAs*, which are incorporated into the task specification by making a product of task DFA and prerequisite DFAs.

Specifically, a prerequisite DFA is a tuple $\mathbb{C} = (\mathcal{C}, \mathcal{G}, T^{\mathbb{C}}, \mathcal{C}^*, c_0)$, where $\mathcal{C}$ is the finite set of automaton states, $\mathcal{G}$ is the input alphabet driving the state transition (same as the set of subgoals), $T^{\mathbb{C}} : \mathcal{C} \times \mathcal{G} \to \mathcal{C}$ is the deterministic transition function, $\mathcal{C}^* \subset \mathcal{C}$ is the set of accepting states, and $c_0$ is the initial state. Some example prerequisite DFAs are shown in Figure 2(c). The left DFA illustrates that visiting any green zone requires visiting an orange zone first, while the right DFA shows that, before visiting a blue zone, the sequence of visiting a purple zone followed by a cyan zone must be adhered to.

## 2.3 Problem Formulation

In this work, we aim to uncover the hierarchical structures of subgoals and subsequently learn a policy efficiently that generates trajectories meeting the constraints of the task DFA specification $\mathbb{B}$ and subgoal prerequisites represented by prerequisite DFAs $\mathbb{C}$. These two sets of constraints can be integrated with environmental MDP by formulating a product MDP defined as below.

**Definition 3.2** (Product MDP) A product MDP synchronizes the environmental MDP with the task DFA and the prerequisite DFAs. Denote the task DFA as $\mathbb{B}$ and the $K$ prerequisite DFAs as $\mathbb{C}_1, \mathbb{C}_2, \ldots, \mathbb{C}_K$. Let $\mathcal{M}^C$ represent the product MDP with a state space defined as $\mathcal{S}^C = \mathcal{S} \times \mathcal{B} \times \mathcal{C}_1 \times \cdots \times \mathcal{C}_K$. Policies over this product MDP are given by $\Pi : \mathcal{S}^C \times \mathcal{A} \to \Delta([0, 1])$. The probabilistic transition function for this product MDP is then defined as:

$$T(s', b', c'_1, \ldots, c'_K | s, b, c_1, \ldots, c_K, a) = \begin{cases} T^{\mathcal{M}}(s'|s,a), & a \in \mathcal{A}(s), (*) \\ 0, & \text{otherwise} \end{cases} \tag{1}$$

where the condition $(*)$ refer to equations $b' = T^{\mathbb{B}}(b'|b, L(s')), c'_1 = T^{\mathbb{C}_1}(c'_1|c_1, L(s')),$ $\ldots, c'_K = T^{\mathbb{C}_K}(c'_K|c_K, L(s'))$, representing state transitions over task DFA and prerequisite DFAs. Denote the trajectory of state-action pairs in product MDP as $\tau = ((s_0, a_0, b_0, c_{1,0}, \ldots, c_{K,0}), \ldots)$.

**Definition 3.3** (Trajectory acceptance) A trajectory $\tau$ is defined to be accepted by task DFA $\mathbb{B}$ and prerequisite DFAs $\mathbb{C}$, i.e., $\tau \models_{\mathbb{C}} \mathbb{B}$, if there exist some $(b, c_1, \ldots, c_K) \in \mathcal{B}^* \times \mathcal{C}_1^* \times \ldots \times \mathcal{C}_K^*$ that are visited by $\tau$ infinitely often.

**Definition 3.4** (Constraint satisfaction) We say a policy $\pi \in \Pi$ satisfies constraints of task DFA $\mathbb{B}$ and prerequisite DFAs $\mathbb{C}$ with the probability $\mathbb{P}[\pi \models_{\mathbb{C}} \mathbb{B}] := \mathbb{E}_{\tau \sim \mathcal{M}_{\pi}^C}[\mathbb{1}_{\tau \models_C \mathbb{B}}]$, where $\mathbb{1}$ indicates whether the trajectory $\tau$ is accepted by $\mathbb{B}$ and $\mathbb{C}$ or not, and $\mathcal{M}_{\pi}^C$ is the distribution of trajectories induced by the policy $\pi$ in product MDP $\mathcal{M}^C$. In addition, we define $\Pi^*$ as the set of policies which maximize the probability of producing trajectories accepted by both $\mathbb{B}$ and $\mathbb{C}$, i.e., $\Pi^* := \{\pi \in \Pi | \mathbb{P}[\pi \models_{\mathbb{C}} \mathbb{B}] = \max_{\pi' \in \Pi} \mathbb{P}[\pi' \models_{\mathbb{C}} \mathbb{B}]\}$. The ultimate goal of this work is to learn a policy in $\Pi^*$ accurately and efficiently.

## 3 Methodology

This work introduces the HDCS framework to address the issues of unknown subgoal structure and sparse reward, including two phases. The diagram is shown in Figure 3. In the first phase (introduced in Section 3.1), hierarchical reinforcement learning (HRL) is employed to uncover a hierarchical structure of subgoals. The first phase is a process of pre-training which makes the agent understand the hidden structure of subgoals in the environment.

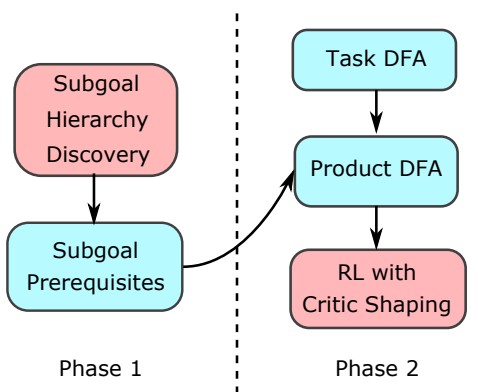

In the second phase (introduced in Section 3.2), to ensure the agent adheres to the subgoal prerequisites in the discovered hierarchy, whenever the task specification (DFA) is given, these learned prerequisites are first integrated into the task DFA via prerequisite DFAs. Then, to address the sparse reward issue in satisfying the LTL specification, the critic shaping method is proposed. This approach effectively mitigates the sparse reward challenge in a simple way, particularly in tasks where the hierarchy of subgoal prerequisites significantly complicates the task formula.

### 3.1 Phase 1: Subgoal Hierarchy Discovery

Figure 3: Diagram of the HDCS framework

In this section, we will first present the HRL-based algorithm for hierarchy discovery, and then discuss how to incorporate learned subgoal prerequisites into the task specification.

### 3.1.1 Discovering Subgoal Hierarchy

We propose to use hierarchical reinforcement learning (HRL) (Dietterich et al., 1998; Nachum et al., 2018) to learn prerequisites of subgoals, which are pre-conditions of visiting subgoals, and use them to build the subgoal hierarchy.

**Model Architecture.** The high-level part is responsible for exploring and learning prerequisites of subgoals. It is realized by an RNN-based RL agent $\pi_\theta^h$ which selects next subgoal from lower levels to explore or achieve a designated subgoal in the current level. The high-level action space is the set of lower-level subgoals. The low-level part is a goal-conditioned RL agent $\pi_\theta^l$ which is trained to achieve any level-0 subgoal (i.e. $\mathcal{G}_0$) selected by the high-level part. The low-level action space is the primitive action space of the environment.

**Discovery Method.** The HRL method discovers subgoal prerequisites level-by-level. The discovery starts from level 0, and proceeds to higher levels incrementally. In level 0, every subgoal $g \in \mathcal{G}_0$ does not have any prerequisites and the low-level policy $\pi_\theta^l(\cdot|g)$ is trained to achieve $g$ by using goal-conditioned RL algorithm (Nachum et al., 2018). In level $l \geq 1$, the operations are divided into exploration phase and learning phase. The exploration phase aims to explore and discover subgoals of the current level, while the learning phase is to learn prerequisites of discovered subgoals. The discovery continues until the prerequisites of all the subgoals in $\mathcal{G}$ of the environment are learned and the subgoal hierarchy is well established. The exploration and learning phases in level $k$ are introduced with details as below. Both $\pi_\theta^l$ and $\pi_\theta^h$ are trained with PPO algorithm (Schulman et al., 2017). The detailed algorithm is shown in Appendix C.

**Exploration Phase.** In each level $k$, the HRL model first works in the exploration phase to explore subgoals of level $k$. The high-level part of the HRL model uses a stochastic policy to select next lower-level subgoal to achieve, and the low-level agent uses its policy to achieve the selected subgoal. In the beginning of exploration phase, the high-level policy $\pi_\theta^h$ is re-initialized while the low-level part $\pi_\theta^l$ keeps the same. In each episode, whenever a new subgoal not in lower levels is achieved, the episode ends and this new subgoal is recorded as a newly discovered subgoal in the level $k$. Then $\pi_\theta^h$ is trained with the reward signal of discovering a new subgoal. The agent will start a new episode until no new subgoals have been discovered for a number of episodes. Then, these new subgoals form the subgoal set of level $k$, i.e., $\mathcal{G}_k$, and the algorithm will proceed to the learning phase.

**Remark.** Note that in level $k$, whenever any subgoal already in $\mathcal{G}_k$ is accidentally visited in exploration phase, this episode ends and this trajectory data is discarded. This is to avoid visiting any subgoals in levels higher than $k$.

**Learning Phase.** In this phase, the agent will learn the prerequisites for subgoals in $\mathcal{G}_k$ one-by-one. For learning each $g \in \mathcal{G}_k$, the high-level part $\pi_\theta^h$ is first re-initialized. Then, being conditioned on $g$, the RNN-based high-level policy $\pi_\theta^h(\cdot|g)$ is used to collect trajectories and trained with the reward signal of achieving $g$, where $\pi_\theta^l$ is fixed and used to achieve the subgoal produced by $\pi_\theta^h(\cdot|g)$. After $\pi_\theta^h(\cdot|g)$ is well trained, the prerequisites of achieving $g$ can be obtained as the output of $\pi_\theta^h(\cdot|g)$, which are sequences of subgoals in levels lower than $k$.

Whenever the prerequisites of subgoals in $\mathcal{G}_k$ are all well learned, the algorithm will proceed to level $k+1$. If the prerequisites of every subgoal in $\mathcal{G}$ are discovered, the algorithm terminates and the subgoal hierarchy is built.

**Remark.** The correctness of learned subgoal sequences for prerequisite can be verified easily via interacting with the environment. After the prerequisite of a target subgoal is learned, the obtained subgoal sequences can be applied into the environment to check whether the target subgoal can be achieved or not. If not, the agent needs to learn the sequences for prerequisite again. However, the empirical experience is that it is very unlikely that the learned subgoal sequences are not correct, since in the learning phase, the high-level policy is trained with the reward of achieving target subgoal only.

## 3.2 Phase 2: Policy Learning for Automaton Satisfaction

In the second stage of HDCS framework, when the task specification (DFA) is given, the agent first incorporates the learned subgoal prerequisites into the task DFA, producing a product DFA $\mathbb{A}_\times$, then the agent learns

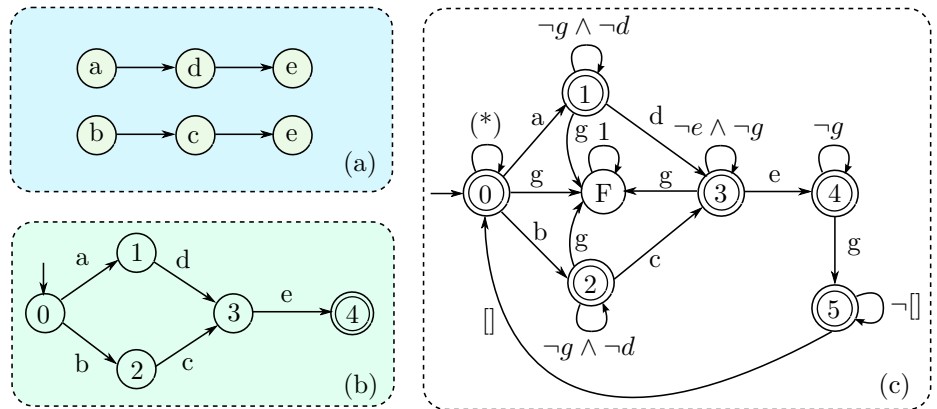

Figure 4: Example of building the prerequisite DFA for subgoal $g$. $a, b, c, d$ are lower-level subgoals. (a): prerequisite sequences of $g$ learned by the HRL method. (b): intermediate DFA. (c): prerequisite DFA. State "F" denotes sink state. The condition (*) over state 1 is $\neg g \wedge \neg a \wedge \neg b$.

an optimal policy to satisfy $\mathbb{A}_\times$, i.e., to achieve an accepting state in $\mathbb{A}_\times$ as often as possible. A thorough discussion on the motivation of critic shaping is presented in Appendix F.

### 3.2.1 Incorporating Subgoal Prerequisites

In the subgoal hierarchy, the prerequisites of subgoals are composed by sequences of lower-level subgoals, which are learned by the HRL method in Section 3.1.1. These prerequisites are incorporated into the task specification via prerequisite DFAs, which consists of two steps.

First, for each target subgoal $g$ we construct an intermediate DFA $\tilde{\mathbb{C}}$ (De la Higuera, 2010; Watanabe et al., 2021) by using its prerequisite sequences of lower-level subgoals. This $\tilde{\mathbb{C}}$ represents all the possible paths for achieving $g$. In $\tilde{\mathbb{C}}$, there is one initial state and one accepting state denoting the achievement of $g$. An example is shown in Figure 4. For the target subgoal $g$, its learned prerequisite sequences are $a \rightarrow d \rightarrow e$ and $b \rightarrow c \rightarrow e$, denoting two *alternative* paths to achieve $g$. Then, these sequences are converted into an intermediate DFA.

Second, we convert the intermediate DFA $\tilde{\mathbb{C}}$ into the prerequisite DFA $\mathbb{C}$. Specifically, this process consists of three steps:

- First, one sink (rejecting) state is added into $\tilde{\mathbb{C}}$ to avoid selecting the target subgoal $g$ before its prerequisites are satisfied, which is the state "F" in Figure 4(c).

- Then, in order to consider the situation that the agent does not need to satisfy $g$ in task completion, except the sink state, all the other states in $\tilde{\mathbb{C}}$ become accepting states, where the prerequisite of $g$ are partially satisfied.

- Finally, one more accepting state is added to represent the situation that $g$ is achieved with its prerequisites satisfied, which is the state 5 in Figure 4(c) as an example. This state is also connected back to the initial state with an edge label of empty ($[]$), meaning that after $g$ is achieved, if the agent wants to achieve it again, its prerequisites have to be satisfied again from the scratch.

**Product DFA.** After prerequisite DFAs of all the subgoals are obtained, we propose to build a product DFA to incorporate the subgoal prerequisite into the task specification. By multiplying the task DFA ($\mathbb{B}$) and all the prerequisite DFAs ($\mathbb{C}_1, \ldots, \mathbb{C}_K$) together, the product DFA is a tuple $\mathbb{A}_\times := (\mathcal{S}_\times, \mathcal{A}_\times, T^\times, \mathcal{S}_\times^*, s_0^\times)$, where the product automaton state space is $\mathcal{S}_\times := \mathcal{B} \times \mathcal{C}_1 \times \ldots \times \mathcal{C}_K$, the input alphabet driving the state transition over $\mathbb{A}_\times$ is $\mathcal{A}_\times := \Sigma \times \mathcal{G} \times \ldots \times \mathcal{G}$, the state transition is $T^\times : \mathcal{S}_\times \times \mathcal{A}_\times \rightarrow \mathcal{S}_\times$, the set of accepting states is $\mathcal{S}_\times^* := \mathcal{B}^* \times \mathcal{C}_1^* \times \ldots \times \mathcal{C}_K^*$, and $s_0^\times$ is the initial state over $\mathbb{A}_\times$. An accepting state in $\mathbb{A}_\times$ is a vector of accepting states in task DFA and all the prerequisite DFAs, making the RL agent to learn to complete the task and obey subgoal prerequisites at the same time. An example of product automaton is shown in Appendix E.

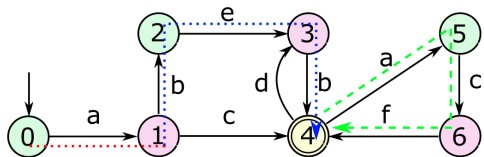

Figure 5: Example of level, stage and round. The number labeled over every state is its index. Yellow state: level 0. Purple state: level 1. Green state: level 2. Red line: stage 2. Blue line: stage 1. Dotted line: round 1. Dashed line: round 2. Self-loops are omitted.

### 3.2.2 Auxiliary Rewards

Due to the complexity of product DFA $\mathbb{A}_\times$ as constraint, the RL agent may suffer from the issue of sparse reward. To overcome this issue, we propose a critic shaping method to accelerate the policy learning under the constraint of an automaton. Different from previous reward shaping methods which add auxiliary rewards onto the reward function, the proposed method directly shapes the critic function by using an max operator, encouraging the agent to reach an accepting state in $\mathbb{A}_\times$ more efficiently. We will first introduce the auxiliary rewards, details of critic shaping operations and finally present its motivation and intuition.

We design auxiliary rewards as intermediate rewards assigned to agent as it makes progress towards the satisfaction of product automaton $\mathbb{A}_\times$. Then, we first define concepts of level, stage and round which are used to formulate the auxiliary rewards $R_A$ for critic shaping.

- **Level.** First, we divide the states of $\mathbb{A}_\times$ into different levels according to their shortest distances to the nearest accepting state in $\mathbb{A}_\times$. The *level k* only contains states with distance $k$. An example of state levels is shown in Figure 5, where the states in the same level have the same color.

- **Stage.** Then, the *stage t* is defined as the segment of agent's trajectory between the *first* visit to level $t$ and *first* visit to level $t-1$. *The stage essentially divides the trajectory according to the shortest distance to an accepting state in $\mathbb{A}_\times$ the agent has ever had in this episode.* As shown in Figure 5, before visiting state 4, the stage 1 and 2 are shown in blue and red lines, respectively. Although state 2 is in level 2, the visit to state 2 is already in stage 1, since the state 1 in level 1 was previously visited and the agent already entered stage 1 at that time.

- **Round.** In addition, the satisfaction of $\mathbb{A}_\times$ means infinitely many visits to an accepting state in $\mathbb{A}_\times$. Hence, we define *round i* as the segment of a trajectory from $(i-1)$-th to $i$-th visit to an accepting state or the end of episode (i.e., time limit is exceeded).

Now we introduce auxiliary rewards $R_A$ defined in terms of stage and round indices. In any round $i$, whenever the agent enters a stage $k$, there will be an auxiliary reward $R_A(i,k) > 0$ to shape the critic function, encouraging the agent making progress towards visiting an accepting state. Whenever the agent visits any accepting state in $\mathbb{A}_\times$ for the $i$-th time (i.e., the round $i$ ends), it will receive an auxiliary reward $R_A(i,0) > 0$ and then start the next round $i+1$. These auxiliary rewards must satisfy an condition as below, for any $i, k$ integers such that $i > 0$ and $k > 0$,

$$R_A(i,k) < \gamma^H R_A(i,k-1), \quad R_A(i,0) < \gamma^H R_A(i+1,k) \tag{2}$$

where $\gamma$ is the discount factor of environmental MDP. *Note that H is the max length of each stage, rather than the whole episode, which allows the agent to visit accepting states in $\mathbb{A}_\times$ infinitely often.* The intuition behind the condition equation 2 will be introduced in Appendix F.

### 3.2.3 Critic Shaping

In order to model the multi-stage nature of satisfying DFA constraint, we extend the state representation to a tuple of four elements, including $s$ (environmental state), $s_\times$ (state in DFA constraint), $i$ (round index),

and $k$ (stage index), which are inputs to policy ($\pi_\theta$) and the critic function ($V_\pi^\theta$). The auxiliary rewards $R_A$ are used only at the boundaries of stages and rounds. Inspired by the max-reward RL (Veviurko et al., 2024), the target for updating critic function $V_\pi^\theta$ is written as below based on Bellman-like equations: whenever the agent steps into stage $k$ from stage $k-1$ in round $i$

$$V_\pi(s_t, s_t^\times, i, k) = \mathbb{E}_{s_{t+1}, a_t \sim \pi}[R_A(i, k-1) \vee \gamma V_\pi(s, s_{t+1}^\times, i, k-1)|s = s_{t+1}] \tag{3}$$

where $s_{t+1}^\times$ is the next state of the transition driven by symbol $L(s_{t+1})$ over $\mathbb{A}_\times$, and $a \vee b$ is $\max(a, b)$. If the agent steps into a new round $i + 1$, the target for updating $V_\pi^\theta$ is computed as below

$$V_\pi(s_t, s_t^\times, i, 0) = \mathbb{E}_{s_{t+1}, a_t \sim \pi}[R_A(i+1, k') \vee \gamma V_\pi(s, s_{t+1}^\times, i+1, k')|s = s_{t+1}] \tag{4}$$

where $k'$ is the stage index of next state $s_{t+1}^\times$ over $\mathbb{A}_\times$. Otherwise, whenever $s_{t+1}^\times$ and $s_t^\times$ are in the same stage and round over $\mathbb{A}_\times$, it is

$$V_\pi(s_t, s_t^\times, i, k) = \mathbb{E}_{s_{t+1}, a_t \sim \pi}[\gamma V_\pi(s, s_{t+1}^\times, i, k)|s = s_{t+1}] \tag{5}$$

In implementation, the policy $\pi_\theta$ and critic $V_\pi^\theta$ are trained with auxiliary rewards by using PPO (Schulman et al., 2017) or DDPG algorithm (Lillicrap et al., 2015). The motivation of critic shaping is introduced in Appendix F, and the convergence proof is presented in Appendix D.

### 3.2.4   Adaptive Design of Auxiliary Reward

Since the agent has little knowledge about the environment initially, any pre-designed auxiliary rewards $R_A$ may not precisely reflect the difficulty of activating desired DFA transitions, leading to sub-optimal policies. In order to address this issue, we propose an adaptive design for $R_A$ which first sets $R_A$ to some initial values in the beginning, and then updates every value of $R_A$ adaptively according to the difficulty of achieving the corresponding stage, which is determined by the empirical performance of achieving that stage in previous episodes.

We first set every $R_A(i, k)$ to its initial value, which is selected empirically for each environment and satisfies the condition equation 2. For example, in Minecraft, we choose $R_A = [5, 10, 20, 40, \ldots]$, while in the AssemblyZone environment, we set initial values of $R_A$ to be $10, 100, 1000, \ldots$

Denote $\beta_{i,k}$ as a binary label indicating whether the stage $k$ in round $i$ is achieved in the last episode or not. If yes, $\beta_{i,k} = 0$. If not, $\beta_{i,k} = 1$. Then, at the end of $l$-th episode, $R_A(i, k)$ is updated as

$$R_A^l(i, k) \leftarrow \left[R_A^{l-1}(i, k) + K_I \beta_{i,k}\right]_{(R_A^{\min}(i,k), R_A^{\max}(i,k))} \tag{6}$$

where $K_I$ is a hyper-parameter controlling the update rate of $R_A$, and $[\cdot]_{(a,b)}$ is the operator clipping a value into the interval $(a, b)$. In experiments, we select $K_I = 1$. Besides, we set $R_A^{\min}(i, k) := \gamma^{-H} R_A^{l-1}(i, k+1)$ and $R_A^{\max}(i, k) := \gamma^H R_A^{l-1}(i, k-1)$, which is to make sure that the condition equation 2 still holds after $l$-th update of $R_A$. This dynamic and adaptive update can make $R_A$ have higher values in difficult stage, giving the agent more encouragement to achieve this kind of stages.

## 4   Experiments

The experiments will try to examine two questions as below:

1. Can the proposed framework accurately learn the prerequisites and build the hierarchy of subgoals?

2. Can the critic shaping method accelerate the solving of LTL task with and without subgoal hierarchy?

In this section, we will use experimental results and analysis to answer these two questions in 4.1 and 4.2, respectively. The environments are introduced in Appendix G. The tasks used in experiments are described in Appendix H. The evaluation of critic shaping in Button domain is in Section 4.3. Ablation study for the whole framework is presented in Appendix K.

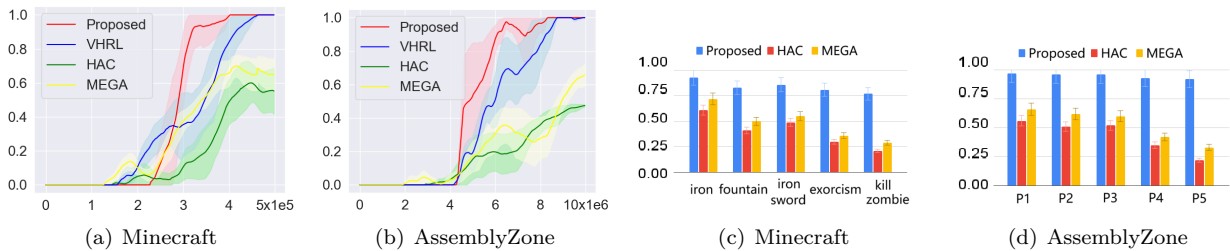

Figure 6: Comparison of exploration. The x-axis is the number of samples drawn from the environment. The y-axis are success rates of reaching target subgoals (a, b) and intermediate subgoals (c, d).

## 4.1 Hierarchy Discovery Evaluation

The baselines used in the evaluation of subgoal hierarchy discovery are presented in Appendix I.1.

In environments with hard-to-explore hierarchical structures, we first conduct experiments to compare the exploration capability of HDCS and state-of-the-art baselines. Then, we compare the accuracy of subgoal hierarchies learned by HDCS and baselines. In addition, more detailed results are presented in Appendix J.

**Exploration.** In 2D-Minecraft and AssemblyZone, we evaluated the learning efficiency of the HDCS framework against baseline methods for attaining the highest-level subgoals (specifically, "exorcism" and "kill zombie" in Minecraft, and $p4$ and $p5$ in AssemblyZone), whose success rates are compared in Figure 6 (a,b). In addition, some selected intermediate subgoals which are used as selected exploration milestones. The success rates of achieving intermediate subgoals are shown in Figure 6 (c,d), where the performances are evaluated after training with $3e5$ and $6e6$ environmental steps. As illustrated in Figure 6, the proposed framework exhibits accelerated learning and surpasses all baselines in both efficiency and success rate. This improvement is primarily due to its level-by-level acquisition of subgoal prerequisites, in accordance with the underlying hidden hierarchy. Although the VHRL approach eventually achieves stable attainment of the target subgoals, it expends an excessive number of samples on intermediate subgoals because it directly targets the highest-level subgoals, thereby neglecting the hierarchical structure. Moreover, both HAC and MEGA implement goal-conditioned high-level policies to directly reach various subgoals without considering the subgoal hierarchy, which ultimately leads to suboptimal performance in achieving the target subgoals with satisfactory sample efficiency.

**Hierarchical Structure Discovery.** In our experiments, the accuracy of subgoal hierarchy discovery is evaluated by computing the Hamming distance between the learned subgoal prerequisites and the true ones. Figure 7 illustrates a comparative analysis between the proposed framework and baseline methods, revealing that our approach significantly outperforms both HAC and MEGA. This superior performance is attributed to the framework's method of sequentially discovering subgoals, thereby progressively constructing a subgoal hierarchy that more accurately reflects the inherent structure of the environment. In contrast, conventional HRL methods like HAC and MEGA rely on exploration-driven subgoal discovery, which overlooks the hierarchical organization of subgoals.

Note that the targets of experiments in Figure 6 and 7 are different. In Figure 6, the experiments are designed to evaluate only the exploration capability of the agent trained by the proposed method. In Figure 7, we compare the proposed and baseline methods for the accuracy of learning prerequisites of all the subgoals. Therefore, the experiments in Figure 7 need more samples than those in Figure 6.

## 4.2 Task Completion Evaluation

The baselines used in task completion evaluation are presented in Appendix I.2.

In addition to the subgoal hierarchy discovery, this work proposes the critic shaping method to accelerate the resolution of the given DFA task, which is the stage 2 of the proposed framework. In this section, we compare the critic shaping method against some state-of-the-art baselines on the sample efficiency of solving DFA tasks. The first set of experiments uses regular tasks without subgoal hierarchy, while the second part of

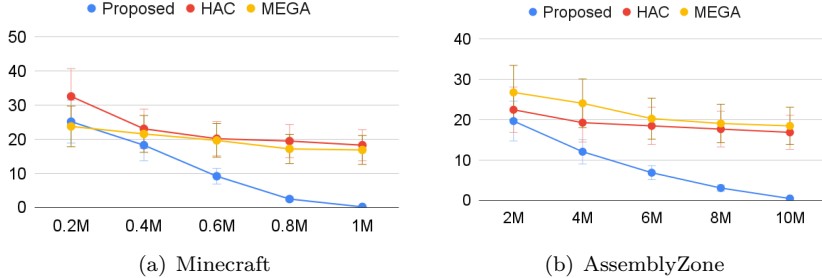

Figure 7: Comparison of accuracy in subgoal hierarchy discovery. The y-axis is the Hamming distance between learned and real subgoal prerequisites.

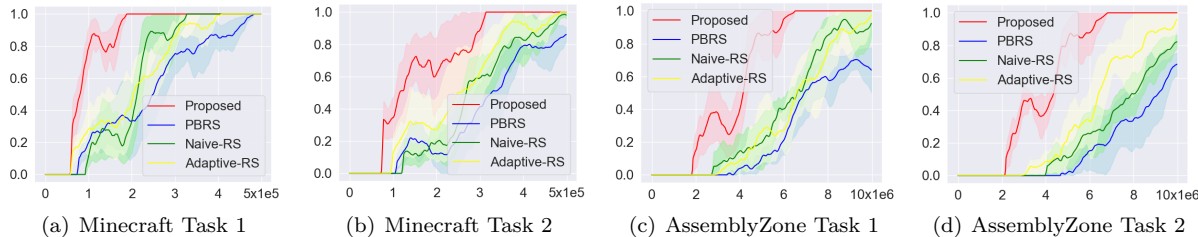

Figure 8: Performance comparison on tasks without subgoal hierarchy. The y-axis is the success rate.

experiments evaluate tasks integrated with subgoal hierarchy, which is the overall evaluation of the proposed framework. The details of evaluated tasks are presented in Appendix H. The evaluation of critic shaping in a more challenging domain is shown in Section 4.3. The ablation study specific to critic shaping is presented in Appendix K

We evaluate the performance using the metric of *success rate* of task completion, calculated by counting the frequency of successful episodes where the task is completed. We pause the learning process every 10K training steps in every domain, then evaluate the current policy in the test environment over 20 episodes. We ran 5 independent trials for each method. In each plot, the solid curve represents the mean performance whose standard deviation is shown by the shaded area.

The comparison on tasks without subgoal hierarchy is shown in Figure 8, where only subgoals in the level 0 are used to compose the tasks. The automata of evaluated tasks are presented in Appendix H. Figure 9 presents the performance comparison on tasks with subgoal hierarchy, where subgoals from every level are used and the subgoal prerequisites learned in the stage 1 of HDCS are incorporated into the task automata via prerequisite DFAs.

As shown in Figures Figure 8 and Figure 9, the proposed framework significantly outperforms baselines, demonstrating the effectiveness of the critic shaping method. Every baseline assign intermediate rewards to the reward function, dependent on the agent's distance towards task completion on the automaton $\mathbb{A}_\times$. Compared with baselines, the critic shaping directly modifies the critic values via a max operator, bypassing

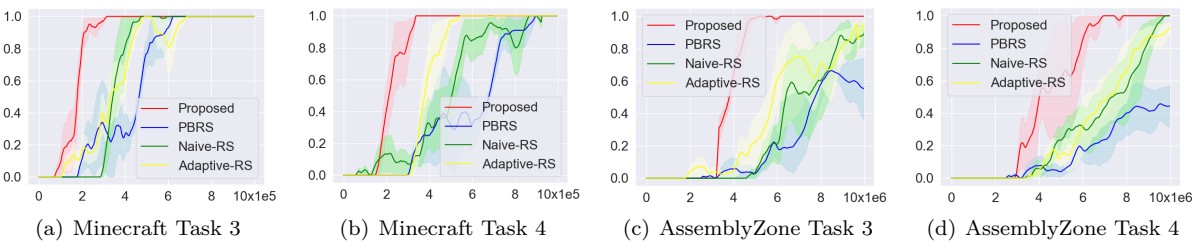

Figure 9: Performance comparison on tasks with subgoal hierarchy. The y-axis is the success rate.

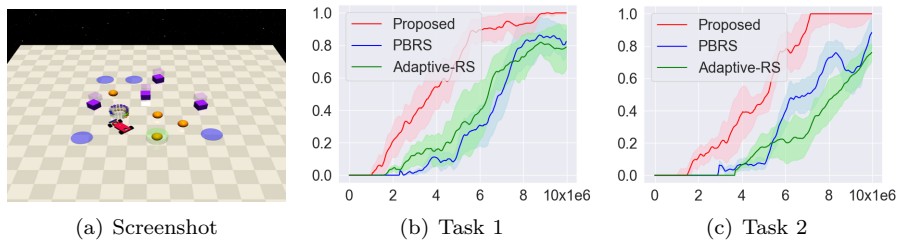

Figure 10: Performance comparison in Button domain.

the process of the critic function that assimilates auxiliary rewards. Besides, the rewards in critic shaping set stable targets for updating the critic function, improving the efficiency of critic training and facilitating the reward propagation across the state space.

The advantage of the HDCS framework is magnified in Figure 9. This is because the DFA constraints there are product automata which incorporated prerequisite DFAs of subgoals, and have more complex structures. Since the agent needs to go through more stages to satisfy more complex DFA constraints, the critic shaping can have larger margin over baselines based on conventional reward shaping. We can see that the PBRS method performs almost worst. This is because the auxiliary rewards on too many automaton states can distract the agent from reaching the real target. Both Naive-RS and Adaptive-RS perform much worse than the proposed framework, and the performance gap is larger in AssemblyZone environment which has a much longer horizon. This shows that the effect of shaping the critic is significant and increase in proportion with the task horizon.

### 4.3 Evaluation of Critic Shaping in Button Domain

In addition, we evaluate the critic shaping on accelerating the resolution of LTL tasks in the Button domain which is more challenging than Zone. This environment is from Safety-gymnasium (Ji et al., 2023), where an agent must press a number of small buttons in a larger space while avoiding cube-shaped 'gremlins' that move in a fixed circular path. A screenshot of this domain is shown in Figure 10(a). The agent is simulated car (Ji et al., 2023). The task specification (DFA) instructs the agent to press some specific buttons in the right orders, while avoiding making contact with gremlins. The symbols in this domain include indices of buttons (i.e., "1", "2", "3" and "4") and "g" for gremlins. The task DFAs are shown in Appendix H.

We use two baselines, i.e. PBRS and Adaptive-RS, introduced in Appendix I.2. In the proposed method, the critic shaping method is adopted and $R_A$ are chosen as $[10, 100, 1000, \ldots]$. The horizon for every stage is 1000 and the discount $\gamma = 0.998$, so that $R_A$ satisfies the condition in equation 2. The proposed method and baselines adopt PPO algorithm, where the clipping $\epsilon$ should be small, e.g., 0.1.

As shown in Figure 10, critic shaping still shows advantages over PBRS and Adaptive-RS in the high-dimensional domain. Compared with comparisons in Minecraft and AssemblyZone domains, the advantage of the proposed method (critic shaping) enlarges here. This is because the agent here is more difficult to control and the gremlin is moving around. In this case, the stabilization of critic targets brought by the critic shaping can accelerate the agent's learning more significantly.

## 5 Conclusion

In this work, we attempt to solve two important issues of completing temporal logic tasks, i.e. unknown subgoal hierarchy and sparse rewards, which are largely ignored or inadequately solved in previous works. First, we propose an HRL-based framework to learn subgoal prerequisites and build a subgoal hierarchy, significantly facilitating the agent's learning process. Second, we propose a simple and novel critic shaping method which can densify the rewards and efficiently accelerate the resolving of the given task. Each component of the proposed framework is evaluated in various domains, verifying its advantages over baselines. In the future, we will consider using large language model (LLM) to extract the subgoal hierarchy from text descriptions of collected trajectories and incorporate it into task DFA.

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

## A  Source Code

The source codes are in the following link: `https://drive.google.com/drive/folders/1YziBp049rzDB7SVtRDh0yupKtUe1qFA5?usp=sharing`

## B  Related Work

We investigate a temporal extension of goal-conditioned reinforcement learning (RL) (Schaul et al., 2015) in which objectives are encoded as automata. Although early work emphasized goals defined solely as prospective target states, subsequent research has broadened the scope to encompass objectives expressed in natural language (Bahdanau et al., 2018; Jiang et al., 2019; Luketina et al., 2019; Sontakke et al., 2023) and temporal logic formulas (Vaezipoor et al., 2021). Conditional deterministic finite automata (cDFAs), while less expressive than free-form language, strike a useful compromise: they retain much of language's user-friendly character yet preserve the precise, unambiguous semantics essential for the growing deployment of AI agents in safety-critical contexts. In addition, the problem of inferring DFAs has been extensively studied (Angluin, 1987; Gold, 1967; Heule & Verwer, 2010; Lauffer et al., 2022; Oncina et al., 1992; Parekh & Honavar, 2001; Wieczorek, 2017), and recent advances demonstrate that both DFAs and cDFAs can be learned directly from natural-language descriptions and expert demonstrations, further narrowing the gap between language-based and automata-based goal specifications (Vazquez-Chanlatte, 2022; Vazquez-Chanlatte et al., 2024).

On the LTL-constrained policy optimization side, several works have tackled the problem in varying settings (Cai et al., 2021; Ding et al., 2014; Shah et al., 2024; Voloshin et al., 2023). Adjacent to these efforts, there is a surge of recent works which have explored LTL specifications and automaton-like models as objectives in RL (Camacho et al., 2019; Fu & Topcu, 2014; Hasanbeig et al., 2018; Sadigh et al., 2014; Wang et al., 2020; Vaezipoor et al., 2021; Alur et al., 2022; De Giacomo et al., 2020; Voloshin et al., 2023; Jothimurugan et al., 2019; 2021). However, none of them tackled the issue of unknown subgoal hierarchy.

Several previous works have tried to address the issue of sparse reward in solving the LTL task. Some of them applied potential-based reward shaping (PBRS) (Ng et al., 1999) to derive auxiliary rewards for intermediate progress towards task completion (Grześ, 2017; Jiang et al., 2021; Devidze et al., 2022; Icarte et al., 2022). However, this method still does not resolve the sparse reward issue adequately. Another method is to decomposing LTL task and solve it by hierarchical RL (HRL) approaches (Araki et al., 2021; Camacho et al., 2019; Icarte et al., 2022). However, these HRL-based methods may suffer from the sub-optimality issues.

## C  Subgoal Hierarchy Discovery Algorithm

The algorithm for discovering subgoal hierarchy and prerequisites is shown in Algorithm 1.

**Remark.** The low-level policy $\pi_\theta^l(\cdot|g)$ is pre-trained to achieve every subgoal $g$ in level 0 ($\mathcal{G}_0$). In pre-training, conditioned on every $g \in \mathcal{G}_0$, $\pi_\theta^l$ is trained by the PPO algorithm to reach $g$. Since every $g \in \mathcal{G}_0$ can be visited without prerequisites and the set $\mathcal{G}_0$ is known to the agent initially, the agent could use $\pi_\theta^l$ to achieve every $g$ directly. For any subgoal in higher levels (i.e., $\tilde{\mathcal{G}}$), the agent could leverage the subgoal hierarchy to decompose it into a sequence of subgoals in level 0, and then apply $\pi_\theta^l$ to achieve these decomposed subgoals one-by-one, meeting the prerequisites.

## D  Convergence Proof of Critic Shaping

In this section, we provide the convergence proof of the critic shaping method. We will first formulate the DFA satisfaction based on critic shaping as reward maximization in an extended MDP, and then prove the convergence of max-Bellman operators used in this extended MDP, implying the convergence of critic shaping method.

---

**Algorithm 1** Subgoal Hierarchy Discovery

---

**Require:** subgoal set $\mathcal{G}$; goal-conditioned low-level policy $\pi_\theta^l(\cdot|g)$; empty subgoal hierarchy $\mathcal{H}$;

1: $\tilde{\mathcal{G}} := \mathcal{G} \setminus \mathcal{G}_0$; $l = 0$; $\mathcal{H} = $ ;
2: **while** $\tilde{G}$ is not empty **do**
3:      $l \leftarrow l + 1$;
4:      initialize high-level policy $\pi_\theta^h$; # *exploration phase starts*
5:      $\mathcal{G}_l \leftarrow []$;
6:      **while** true **do**
7:          explore until a new subgoal is discovered;
8:          **if** no more new subgoal has been discovered for many episodes **then**
9:              break;
10:          **end if**
11:          store the new subgoal into $\mathcal{G}_l$;
12:          train high-level policy;
13:      **end while**
14:      # *exploitation phase starts*
15:      **for** each $g \in \mathcal{G}_l$ **do**
16:          initialize high-level policy $\pi_\theta^h$;
17:          train $\pi_\theta^h(\cdot|g)$ to reach $g$;
18:          obtain prerequisites of $g$ from the trained; $\pi_\theta^h(\cdot|g)$ and store them into $\mathcal{H}$
19:          remove $g$ from $\tilde{\mathcal{G}}$;
20:      **end for**
21: **end while**

---

## D.1 Extended MDP

In the formulation in Section 3.2.3, the state representation is extended to a tuple of four elements, including $s$ (environmental state), $s_\times$ (state of product DFA $\mathbb{A}_\times$), $i$ (round index) and $k$ (stage index). To formalize this extension, we define the *extended state space* $\hat{\mathcal{S}} := \mathcal{S} \times \mathcal{S}_\times \times \mathbb{Z} \times \mathbb{Z}$, where $\mathcal{S}$ is the environmental state space, $\mathcal{S}_\times$ is the set of automaton states on the product DFA $\mathbb{A}_\times$ and $\mathbb{Z}$ denotes the set of integers. Besides, we also define the level function $l : \mathcal{S}_\times \to \mathbb{Z}$ mapping an automaton state on $\mathbb{A}_\times$ to its level. As defined in Section 3.2.2, the level of a state means the minimal number of hops to an accepting state over $\mathcal{A}_\times$).

For any extended state $\hat{s} = (s, s_\times, k, i) \in \hat{\mathcal{S}}$, the initial distribution of extended states is defined as $\hat{p}_0(\hat{s}) := p_0(s)\delta(s_\times)\delta(k - l(s_\times))\delta(i)$, where $p_0$ is the initial state distribution of environmental MDP, $\delta(s_\times)$ ensures that the agent starts from the initial state 0 of $\mathbb{A}_\times$, and other $\delta$ terms ensure that the first stage index is the level of automaton state 0 and the round index starts from 0.

Then, for any extended state $\hat{s} \in \hat{\mathcal{S}}$ and for an action $a \in \mathcal{A}$, the *extended transition function* $\hat{P}(\cdot, \cdot, \cdot, \cdot|s, s_\times, k, i, a)$ is a PDF over $(s', s'_\times, k', i') \in \hat{\mathcal{S}}$, defined as

$$\hat{P}(s', s'_\times, k', i'|s, s_\times, k, i, a) := P(s'|s, a) \cdot \delta(s'_\times - T^\times(s_\times, L(s'))) \cdot \delta(k' - l(s'_\times) \wedge k) \cdot \delta(i' - (i + \mathbf{1}\{s'_\times \in \mathcal{S}^*_\times\})) \quad (7)$$

where $P$ is the transition function of the environmental MDP, $T^\times$ is the state transition function of $\mathbb{A}_\times$, $a \wedge b$ denotes $\min(a, b)$, and $\mathcal{S}^*_\times$ is the set of accepting states of $\mathbb{A}_\times$. The first two terms in equation 7 denote the state transitions in environmental MDP and $\mathbb{A}_\times$. The third term means that the stage index $k$ is the minimal level the agent has reached in the current round. The last term denotes that the round index $i$ is the number of accepting visits the agent has made in the current episode. Since we only consider deterministic environment in this work, $P$ becomes a delta function and hence the distribution produced by $\hat{P}$ concentrates on a single point.

In addition, the auxiliary reward function $R_A$ is defined in Section 3.2.2 and is used to train the policy $\pi : \hat{\mathcal{S}} \to \Delta(\mathcal{A})$ to visit accepting states of $\mathbb{A}_\times$ (*accepting visit*) as often as possible in one episode. Based on

$R_A$, we can define the reward function for extended state transition $(\hat{s}, a, \hat{s}')$ and action $a$ as below

$$\hat{R}(\hat{s}, a, \hat{s}') = \begin{cases} R_A(k', i'), & \text{if } 0 < l(s'_\times) < k \\ R_A(k', i'), & \text{if } s'_\times \in \mathcal{S}^*_\times \\ 0 & \text{otherwise} \end{cases} \tag{8}$$

In the definition of $\hat{R}$, the first line denotes the situation where a new stage is achieved by $\hat{s}'$ and the second line represents the case when an accepting state is visited in $\hat{s}'$. Combining all the discussions above, we can have the following definition.

**Extended MDP.** The extended MDP is an MDP $\hat{M}$ defined by the tuple $(\hat{\mathcal{S}}, \mathcal{A}, \hat{R}, \hat{P}, \hat{p}_0, \gamma)$. Here $\mathcal{A}$ and $\gamma$ are action space and discount factor of the environmental MDP $M$.

## D.2 Convergence Proof

As introduced in Section 3.2.3, in the proposed critic shaping method, auxiliary rewards $R_A$ are used to directly shape the critic function via the max operator $\vee$. So, for any time step $t$ and any extended state $\hat{s}_t = (s_t, s_{\times,t}, k_t, i_t)$, we can define the critic function $Q$ as below,

$$
\begin{aligned}
Q^\pi(\hat{s}_t, a_t) &= \mathop{\mathbb{E}}_{\substack{\hat{s}_{t+1} \sim \hat{P}(\cdot|s_t, a_t) \\ a_{t+1} \sim \pi(\hat{s}_{t+1})}} \left[ \max\left( \hat{R}(\hat{s}_t, a_t, \hat{s}_{t+1}), \gamma \mathop{\mathbb{E}}_{\substack{\hat{s}_{t+2} \sim \hat{P}(\cdot|s_{t+1}, a_{t+1}) \\ a_{t+2} \sim \pi(\hat{s}_{t+2})}} [\max(\hat{R}(\hat{s}_{t+1}, a_{t+1}, \hat{s}_{t+2}), \ldots)] \right) \right] \quad (9) \\
&= \mathop{\mathbb{E}}_{\substack{\hat{s}_{t+1} \sim \hat{P}(\cdot|s_t, a_t) \\ a_{t+1} \sim \pi(\hat{s}_{t+1})}} \left[ \max\left( \hat{R}(\hat{s}_t, a_t, \hat{s}_{t+1}), Q^\pi(\hat{s}_{t+1}, a_{t+1}) \right) \right] \quad (10)
\end{aligned}
$$

According to the condition equation 2 of $R_A$ values and the definitions of the reward $\hat{R}$ and the Q function in equation 8 and equation 9, the objective of visiting accepting states in the DFA $\mathbb{A}_\times$ as often as possible in a finite episode is equivalent to maximizing the Q value in this extended MDP. So, the optimal policy of satisfying the product DFA $\mathbb{A}_\times$ can be obtained by training a stationary policy to maximize the Q function defined in equation 9. Therefore, the convergence of policy optimization based on this Q function implies the convergence of the critic shaping method.

In order to show the convergence, we first define the max-Bellman evaluation operator and the max-Bellman optimality operator (Quah & Quek, 2006; Veviurko et al., 2024). Based on a critic function $Q : \hat{\mathcal{S}} \times \mathcal{A} \to \mathbb{R}$ and for any state-action pair $(\hat{s}, a) \in \hat{\mathcal{S}} \times \mathcal{A}$,

$$
\begin{aligned}
(\mathcal{M}^\pi Q)(\hat{s}, a) &= \mathbb{E}_{\hat{s}' \sim \hat{P}(\cdot|\hat{s}, a)}[\max(\hat{R}(\hat{s}, a, \hat{s}'), \gamma \mathbb{E}_{a' \sim \pi(\cdot|\hat{s}')}[Q(\hat{s}', a')])] \\
(\mathcal{M}^* Q)(\hat{s}, a) &= \mathbb{E}_{\hat{s}' \sim \hat{P}(\cdot|\hat{s}, a)}[\max(\hat{R}(\hat{s}, a, \hat{s}'), \gamma \max_{a' \in \mathcal{A}}[Q(\hat{s}', a')])]
\end{aligned} \tag{11}
$$

Specifically, $\mathcal{M}^\pi$ and $\mathcal{M}^* : (\hat{\mathcal{S}} \times \mathcal{A} \to \mathbb{R}) \to (\hat{\mathcal{S}} \times \mathcal{A} \to \mathbb{R})$ are operators that take in a Q function and returns another modified Q function by assigning the Q value of the input state-action pair $(s, a)$, to be the maximum of the reward obtained at $(s, a)$ and the discounted future expected Q value. In the following, we are going to show that these two Bellman operators converge to a fixed point, and this fixed point is the optimal policy which maximizes the Q values. The proofs are variants of convergence proofs in max-reward RL (Quah & Quek, 2006; Gottipati et al., 2020).

**Theorem D.1.** *The operator $\mathcal{M}^*$ defined in equation 11 have the following properties.*

1. *Monotonicity: Assume that we have two Q functions $Q_1, Q_2 : \hat{\mathcal{S}} \times \mathcal{A} \to \mathbb{R}$ such that $Q_1(\hat{s}, a) \geq Q_2(\hat{s}, a)$ for any state-action pair $(\hat{s}, a)$. Then*

$$\mathcal{M}^* Q_1 \geq \mathcal{M}^* Q_2 \tag{12}$$

2. *Contraction: both operators are $\gamma$-contraction in the supremum norm ($\|\cdot\|_\infty$), i.e., for any Q functions $Q_1, Q_2 : \hat{\mathcal{S}} \times \mathcal{A} \to \mathbb{R}$, we have*

$$\|\mathcal{M}^* Q_1 - \mathcal{M}^* Q_2\|_\infty \leq \gamma \|Q_1 - Q_2\|_\infty \tag{13}$$

*Proof.* We first prove the monotonicity and contraction for $\mathcal{M}^*$.

**Monotonicity.** Assume $Q_1$ and $Q_2$ are two Q functions such that $Q_1(\hat{s}, a) \geq Q_2(\hat{s}, a)$ for any $(\hat{s}, a) \in \hat{\mathcal{S}} \times \mathcal{A}$. Then, we have

$$\max_{a' \in \mathcal{A}} Q_1(\hat{s}, a') \geq Q_2(\hat{s}, a), \forall (\hat{s}, a) \in \hat{\mathcal{S}} \times \mathcal{A}$$

which yields

$$\max_{a' \in \mathcal{A}} Q_1(\hat{s}, a') \geq \max_{a' \in \mathcal{A}} Q_2(\hat{s}, a'), \forall \hat{s} \in \hat{\mathcal{S}}$$

Based on the definition of operator $\mathcal{M}^*$ in equation 11, we have

$$\mathcal{M}^* Q_1(\hat{s}, a) \geq \gamma \mathbb{E}_{\hat{s}' \sim \hat{P}(\cdot|\hat{s}, a)} \left[ \max_{a' \in \mathcal{A}} Q_1(\hat{s}, a') \right]$$

Then, it is obvious that

$$\mathcal{M}^* Q_1(\hat{s}, a) \geq \gamma \mathbb{E}_{\hat{s}' \sim \hat{P}(\cdot|\hat{s}, a)} \left[ \max_{a' \in \mathcal{A}} Q_2(\hat{s}, a') \right] \tag{14}$$

The definition of $\mathcal{M}^*$ also implies that

$$\mathcal{M}^* Q_1(\hat{s}, a) \geq \mathbb{E}_{\hat{s}' \sim \hat{P}(\cdot|\hat{s}, a)} [\hat{R}(\hat{s}, a, \hat{s}')] \tag{15}$$

Since we only consider deterministic environment in this work (stated in Section 2.1), based on equation 14 and equation 15, we have

$$\mathcal{M}^* Q_1(\hat{s}, a) \geq \mathbb{E}_{\hat{s}' \sim \hat{P}(\cdot|\hat{s}, a)} \left[ \max \left( \hat{R}(\hat{s}, a, \hat{s}'), \gamma \max_{a' \in \mathcal{A}} Q_2(\hat{s}', a') \right) \right] = \mathcal{M}^* Q_2(\hat{s}, a) \tag{16}$$

**Contraction.** Denote $f_i(\hat{s}, a, \hat{s}') := \gamma \max_{a' \in \mathcal{A}} Q_i(\hat{s}', a')$. Based on the fact that $\max(x, y) = 0.5(x + y + |x - y|), \forall x, y \in \mathbb{R}$, we have

$$
\begin{aligned}
\max(\hat{R}, f_1) - \max(\hat{R}, f_2) &= 0.5(\hat{R} + f_1 + |\hat{R} - f_1|) - 0.5(\hat{R} + f_2 + |\hat{R} - f_2|) \\
&= 0.5(f_1 - f_2 + |\hat{R} - f_1| - |\hat{R} - f_2|) \\
&\leq 0.5(f_1 - f_2 + |\hat{R} - f_1 - (\hat{R} - f_2)|) \\
&= 0.5(f_1 - f_2 + |f_1 - f_2|) \\
&\leq |f_1 - f_2|
\end{aligned}
\tag{17}
$$

Then, we can have

$$
\begin{aligned}
\|\mathcal{M}^* Q_1 - \mathcal{M}^* Q_2\|_\infty &= \|\mathbb{E}_{\hat{s}' \sim \hat{P}(\cdot|\hat{s}, a)}[\max(\hat{R}, f_1) - \max(\hat{R}, f_2)]\|_\infty \\
&\leq \|\mathbb{E}_{\hat{s}' \sim \hat{P}(\cdot|\hat{s}, a)}[f_1 - f_2]\|_\infty \\
&= \|\mathbb{E}_{\hat{s}' \sim \hat{P}(\cdot|\hat{s}, a)}[\max_{a' \in \mathcal{A}} Q_1(\hat{s}', a') - \max_{a' \in \mathcal{A}} Q_2(\hat{s}', a')]\|_\infty \\
&\leq \|Q_1 - Q_2\|_\infty
\end{aligned}
$$

which proves the $\gamma$-contraction of $\mathcal{M}^*$ in equation 13. The left-hand side of this equation represents the largest difference between the two Q-functions. Since $\gamma$ is in the range $[0, 1)$, all the differences are guaranteed to converge to zero in the limit. Based on the Banach fixed point theorem, the operator $\mathcal{M}^*$ admits a fixed point. □

**Theorem D.2.** *The operator $\mathcal{M}^\pi$ defined in equation 11 have the following properties.*

1. *Monotonicity: Assume that we have two Q functions $Q_1, Q_2 : \hat{\mathcal{S}} \times \mathcal{A} \to \mathbb{R}$ such that $Q_1(\hat{s}, a) \geq Q_2(\hat{s}, a)$ for any state-action pair $(\hat{s}, a)$. Then*

$$\mathcal{M}^\pi Q_1 \geq \mathcal{M}^\pi Q_2 \tag{18}$$

2. *Contraction: both operators are $\gamma$-contraction in the supremum norm ($\|\cdot\|_\infty$), i.e., for any Q functions $Q_1, Q_2 : \hat{\mathcal{S}} \times \mathcal{A} \to \mathbb{R}$, we have*

$$\|\mathcal{M}^\pi Q_1 - \mathcal{M}^\pi Q_2\|_\infty \leq \gamma \|Q_1 - Q_2\|_\infty \tag{19}$$

*Proof.* Now we prove the monotonicity and contraction for $\mathcal{M}^\pi$.

**Monotonicity.** Assume $Q_1$ and $Q_2$ are two Q functions such that $Q_1(\hat{s}, a) \geq Q_2(\hat{s}, a)$ for any $(\hat{s}, a) \in \hat{\mathcal{S}} \times \mathcal{A}$. Then, we have

$$\gamma \mathbb{E}_{\hat{s}' \sim \hat{P}(\cdot|\hat{s},a), a' \sim \pi(\cdot|\hat{s}')}[Q_1(\hat{s}', a')] \geq \gamma \mathbb{E}_{\hat{s}' \sim \hat{P}(\cdot|\hat{s},a), a' \sim \pi(\cdot|\hat{s}')}[Q_2(\hat{s}', a')], \forall (\hat{s}, a) \in \hat{\mathcal{S}} \times \mathcal{A}$$

Based on the definition of operator $\mathcal{M}^\pi$ in equation 11, we have

$$\mathcal{M}^\pi Q_1(\hat{s}, a) \geq \gamma \mathbb{E}_{\hat{s}' \sim \hat{P}(\cdot|\hat{s},a), a' \sim \pi(\cdot|\hat{s}')}[Q_1(\hat{s}', a')]$$

Then, it is obvious that

$$\mathcal{M}^\pi Q_1(\hat{s}, a) \geq \gamma \mathbb{E}_{\hat{s}' \sim \hat{P}(\cdot|\hat{s},a), a' \sim \pi(\cdot|\hat{s}')}[Q_2(\hat{s}', a')] \tag{20}$$

The definition of $\mathcal{M}^\pi$ also implies that

$$\mathcal{M}^\pi Q_1(\hat{s}, a) \geq \mathbb{E}_{\hat{s}' \sim \hat{P}(\cdot|\hat{s},a)}[\hat{R}(\hat{s}, a, \hat{s}')] \tag{21}$$

Since we only consider deterministic environment in this work (stated in Section 2.1), based on equation 20 and equation 21, we have

$$\mathcal{M}^\pi Q_1(\hat{s}, a) \geq \mathbb{E}_{\hat{s}' \sim \hat{P}(\cdot|\hat{s},a)}\left[\max\left(\hat{R}(\hat{s}, a, \hat{s}'), \gamma \mathbb{E}_{a' \sim \pi(\cdot|\hat{s}')} Q_2(\hat{s}', a')\right)\right] = \mathcal{M}^* Q_2(\hat{s}, a) \tag{22}$$

which shows the monotonicity of $\mathcal{M}^\pi$.

**Contraction.** Denote $f_i(\hat{s}, a, \hat{s}') := \gamma \mathbb{E}_{a' \sim \pi(\cdot|\hat{s}')} Q_i(\hat{s}', a')$. Based on the fact that $\max(x, y) = 0.5(x + y + |x - y|), \forall x, y \in \mathbb{R}$, we have

$$
\begin{aligned}
\max(\hat{R}, f_1) - \max(\hat{R}, f_2) &= 0.5(\hat{R} + f_1 + |\hat{R} - f_1|) - 0.5(\hat{R} + f_2 + |\hat{R} - f_2|) \\
&= 0.5(f_1 - f_2 + |\hat{R} - f_1| - |\hat{R} - f_2|) \\
&\leq 0.5(f_1 - f_2 + |\hat{R} - f_1 - (\hat{R} - f_2)|) \\
&= 0.5(f_1 - f_2 + |f_1 - f_2|) \\
&\leq |f_1 - f_2|
\end{aligned}
\tag{23}
$$

Then, we can have

$$
\begin{aligned}
\|\mathcal{M}^\pi Q_1 - \mathcal{M}^\pi Q_2\|_\infty &= \|\mathbb{E}_{\hat{s}' \sim \hat{P}(\cdot|\hat{s},a)}[\max(\hat{R}, f_1) - \max(\hat{R}, f_2)]\|_\infty \\
&\leq \|\mathbb{E}_{\hat{s}' \sim \hat{P}(\cdot|\hat{s},a)}[f_1 - f_2]\|_\infty \\
&= \|\mathbb{E}_{\hat{s}' \sim \hat{P}(\cdot|\hat{s},a)}[\mathbb{E}_{a' \sim \pi(\cdot|\hat{s}')} Q_1(\hat{s}', a') - \mathbb{E}_{a' \sim \pi(\cdot|\hat{s}')} Q_2(\hat{s}', a')]\|_\infty \\
&\leq \|Q_1 - Q_2\|_\infty
\end{aligned}
$$

which proves the $\gamma$-contraction of $\mathcal{M}^*$ in equation 19. The left-hand side of this equation represents the largest difference between the two Q-functions. Since $\gamma$ is in the range $[0, 1)$, all the differences are guaranteed to converge to zero in the limit. Based on the Banach fixed point theorem, the operator $\mathcal{M}^\pi$ admits a fixed point. $\square$

According to equation 11 and equation 10, the fixed point of $\mathcal{M}^\pi$ is $Q^\pi$. Denote the fixed point of $\mathcal{M}^*$ operator as $Q^*$. In the following, we will show that $Q^*$ corresponds to the optimal action-value function which maximizes the reward $\hat{R}$ obtained in one episode, i.e., $Q^* = \max_\pi Q^\pi$

**Theorem D.3.** *Define the deterministic policy $\pi^*$ as $\pi^*(\hat{s}) = \arg\max_{a \in \mathcal{A}} Q^*(\hat{s}, a), \forall \hat{s} \in \hat{\mathcal{S}}$. Then, $\pi^*$ is the optimal policy and for any stationary policy $\pi$ and any state-action pair $(\hat{s}, a) \in \hat{\mathcal{S}} \times \mathcal{A}$, we have $Q^{\pi^*}(\hat{s}, a) = Q^*(\hat{s}, a) \geq Q^\pi(\hat{s}, a)$.*

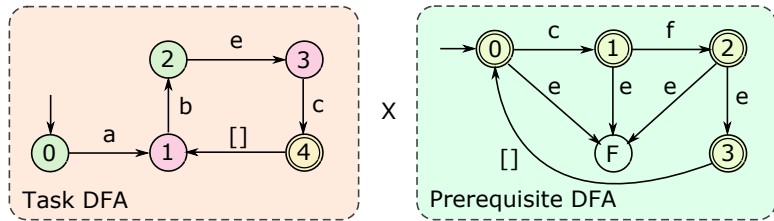

Figure 11: Example of task DFA $\mathbb{B}$ and prerequisite DFA $\mathbb{C}$ for building the product automaton $\mathbb{A}^\times$. "$\times$" denotes the product of two DFAs. Self-loops are omitted.

*Proof.* Based on the definition of $\pi^*$ and $Q^*$, we have $\mathcal{M}^{\pi^*}Q^* = \mathcal{M}^*Q^* = Q^*$. It shows that $Q^*$ is the fixed point of $\mathcal{M}^{\pi^*}$, which implies that $Q^*$ is the value function of $\pi^*$.

For any Q function $Q : \hat{\mathcal{S}} \times \mathcal{A} \to \mathbb{R}$ and any stationary policy $\pi$, we have $\mathcal{M}^*Q \geq \mathcal{M}^\pi Q$. Based on the monotonicity of operators, we have

$$(\mathcal{M}^*)^2 Q \geq \mathcal{M}^*(\mathcal{M}^\pi)Q \geq (\mathcal{M}^\pi)^2 Q$$

Then, by using induction, we can show that for any $n$ we have $(\mathcal{M}^*)^n Q \geq (\mathcal{M}^\pi)^n Q$. Based on the contracting properties of $\mathcal{M}^*$ and $\mathcal{M}^\pi$, both $(\mathcal{M}^*)^n Q$ and $(\mathcal{M}^\pi)^n Q$ converge to $Q^*$ and $Q^\pi$ when $n \to \infty$, respectively. Then, we show that $Q^* \geq Q^\pi$ element-wisely. $\square$

According to theorems above, we show that both the optimality $\mathcal{M}^*$ and evaluation $\mathcal{M}^\pi$ operators, defined based on the max-Bellman operator used in the critic shaping, converge to their fixed points, and the fixed point of $\mathcal{M}^*$ is the optimal policy. Therefore, we show that policy optimization based on the critic shaping method can converge to the optimal policy which maximizes the number of visits to accepting states of $\mathbb{A}_\times$ in a finite episode.

| Levels | States in $\mathbb{A}^\times$ |
|:------:|:-----------------------------:|
| 0 | $(4,0), (4,1), (4,2), (4,3)$ |
| 1 | $(3,0), (3,1), (3,2), (3,3)$ |
| 2 | $(2,2)$ |
| 3 | $(1,2), (2,1)$ |
| 4 | $(2,0), (1,1), (0,2)$ |
| 5 | $(1,0), (0,1), (1,3)$ |
| 6 | $(0,0), (0,3)$ |

Table 1: Dividing states in product automaton $\mathbb{A}^\times$ into levels.

# E  Example of Product Automaton and Levels

Additionally, as shown in Figure 11, we give another example for the levels in a product automaton $\mathbb{A}^\times$. The prerequisite DFA states that the pre-condition of visiting subgoal "e" is visiting "c" first and then "f". The state of product automaton $(s_B, s_C)$ is a tuple of state $s_B$ in task DFA $\mathbb{B}$ and state $s_C$ in prerequisite DFA $\mathbb{C}$. Then, the states in product automaton are divided into different levels as shown in Table 1.

States in level 0 are accepting states in $\mathbb{A}^\times$. States in level $k$ are $k$ hops away from any accepting state following the shortest path over $\mathbb{A}^\times$. Note that the state $(2,3)$ can never be reached from initial state $(0,0)$, since the conditions of transiting from state 2 to 3 coincide in both $\mathbb{B}$ and $\mathbb{C}$, which is visiting "e". States $(0,3)$

and $(1,3)$ are in level 6 and 5 respectively, since the agent goes to these states by visiting "e" redundantly and has to visit "e" by satisfying its causal constraint again.

## F   Discussion: Reward vs Critic Shaping

In comparison to previous reward shaping approaches (Ng et al., 1999; Grześ, 2017; Jiang et al., 2021; Devidze et al., 2022; Icarte et al., 2022), the critic shaping proposed in HDCS offers three primary advantages.

- First, the auxiliary rewards in critic shaping depend solely on the stages and rounds, making them simpler to compute than those in PBRS (Ng et al., 1999) which need a potential function (see Figure 12).

- Second, because the policy is trained using critics rather than rewards, critic shaping directly modifies the critic function, thereby expediting policy learning more effectively than previous reward shaping methods. In prior techniques, auxiliary rewards are first added to the original reward function and then incorporated into the critic function, which can lead to inefficient sample usage and slower policy learning.

- Third, critic shaping provides stable targets for learning the critic function, preventing the agent from being influenced by other subgoals and thus stabilizing the overall policy learning process. This aspect is particularly important for satisfying DFA constraints, as DFAs consist of multiple subgoals, many of which do not have progress toward the DFA satisfaction and can destabilize the agent's learning process. As illustrated in Figure 12, in reward shaping the second visit to state 1 yields an auxiliary reward of $-1.02$ that affects the target for updating critic function at state 2. However, in critic shaping, since the second visit to state 1 there is no progression to a new stage, this visit receives an auxiliary reward of 0, which does not influence the critic target at state 2 and thereby maintains greater stability in the critic and policy learning processes.

**Remark.** Another example of the benefit of critic shaping about stability is shown in Figure 13. When the agent is not well trained, especially in early training period, the critic values in every stage are almost small, as shown by the blue curve in Figure 13. In this case, the high value of $R_A(i, k)$ and the max operator in equation 3 and equation 4 can prevent the agent's learning in stage $k$ from being influenced by the poorly-learned critic values in stage $k - 1$ (blue curve). It essentially inserts a barrier at the boundary between stages $k$ and $k - 1$, and fixes the critic value of target states in stage $k$ at $R_A(i, k)$. This barrier can make the agent's learning of stage $k$ focus on going to stage $k - 1$, stabilizing its learning behavior.

If we simply use "+" in equation 3 and equation 4, $R_A(i, k)$ cannot work like a barrier and the small values in stage $k - 1$ will be added to critic targets in stage $k$, destabilizing and slowing down the agent's learning in stage $k$. Therefore, $R_A(i, k)$ can stabilize the critic's learning in stage $k$ and accelerate the policy learning. This effect can be extended to other stages with other $R_A$.

**Intuition of equation 2.** Due to the discount factor $\gamma$ of the MDP, the auxiliary reward $R_A(i, k)$ can be attenuated exponentially when the critic function is updated according to equation 3 and equation 5. If the trajectory is long, $R_A$ may become small at earlier stages, say stage $k + 1$. If the condition equation 2 is violated, i.e., $\gamma^T R_A(i, k) < R_A(i, k + 1)$ ($T$ denotes the number of time steps from stage $k + 1$ to $k$), the reward information in stage $k$ cannot back-propagate to stage $k + 1$ and earlier stages, which makes the agent's policy learning in stage $k + 1$ be unaware of rewards in stage $k$ and lose optimality. Therefore, we have to make sure that the condition of $R_A$ in equation 2 holds. This condition guarantees that the reward information in every stage and round can back-propagate till the beginning of the trajectory (initial state), which can make the agent's policy learning reach optimality.

The examples of $R_A$ and their discounted values are shown in Figure 13. The solid lines show that values of $R_A$ satisfy the condition equation 2. The red dashed line shows the case where equation 2 is violated, where the red line can block the back-propagation of reward information and make the return estimation at stage $k$ inaccurate, deviating the policy learning from its optimal solution.

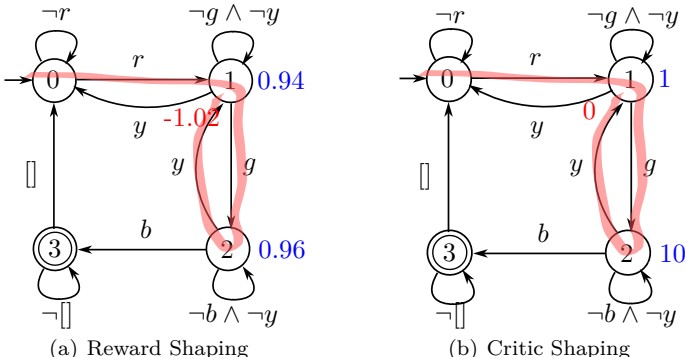

(a) Reward Shaping       (b) Critic Shaping

Figure 12: Comparison of reward and critic shaping. A task DFA in the example environment (Figure 2(a)) is shown in (a) and (b). The red line denotes the agent's path $(r, g, y)$ on the DFA, where state 2 is visited twice by mistake. (a): Assume that the potential function of states are $\phi(0) = -3, \phi(1) = -2, \phi(2) = -1, \phi(3) = 0$ and discount factor $\gamma = 0.98$. Based on PBRS, the auxiliary rewards are computed as $\phi(s') - \gamma \cdot \phi(s)$ and denoted as red and blue numbers. (b): state 1, 2 and 3 are in stage 3, 2 and 1, respectively. The auxiliary rewards in terms of stage are denoted as red and blue numbers. Note that in (a) the critic value of state 2 will be influenced by the second visit to state 1. But in (b), it will be fixed as 10, no matter where the agent goes after visiting state 2 (except state 3 in a better stage).

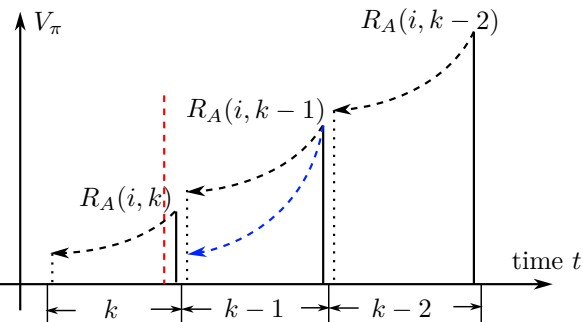

Figure 13: Examples of auxiliary rewards $R_A$. Each $R_A$ is shown as a solid line, which is discounted over one stage in back-propagation, and its discounted value is shown as a dotted line. The dashed arrow denotes the direction of reward back-propagation process. $k, k-1$ and $k-2$ denote stage indices. $i$ is the index of a round. The vertical axis is the value of critic function. The red line denotes a wrong choice of $R_A$ value which blocks the back-propagation of reward information. The blue dashed line denote the situation that the critic values are small in stage $k-1$ when the critic function is not well trained. In this case, the max operator can make the high value of $R_A(i, k)$ a barrier and prevent the agent from being influenced by the critic values in stage $k-1$, stabilizing the agent's learning in stage $k$.

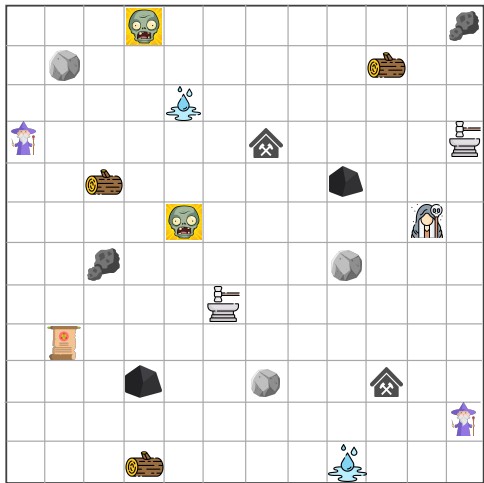

Figure 14: Map of Minecraft. Only subgoals in the level 0 are shown. Higher-level subgoals become visible after their prerequisites are satisfied.

## G   Environments

We evaluate the proposed framework in two domains, including 2D-Minecraft and AssemblyZone. These domains all have complex hierarchical structures of subgoals with very sparse rewards, covering both discrete and continuous state-action spaces. The task is specified by the user in the form of LTL specification. Different from previous papers (Sohn et al., 2018; Hu et al., 2022), we consider the hierarchical subgoal structures containing prerequisites under temporal ordering constraints, which are common in many real-world scenarios. Note that every subgoal has multiple copies in the map, requiring the agent to decide which copy is the optimal to visit.

### G.1   2D-Minecraft

2D-Minecraft is a modified 2D version of the famous Minecraft (Guss et al., 2019). In one episode with limited number of steps, the agent must navigate the map, pick up various materials, and craft tools to complete the task. Compared with original Minecraft game (Guss et al., 2019; Hu et al., 2022), we add some more objects to complicate their relationships. As shown in Figure 14, in a $12 \times 12$ grid-world there are 11 objects corresponding to the subgoals of level 0, including wood, stone, water, scroll, tool workshop, iron workshop, wizard, necromancer, iron ore, coal and zombie (verbs are omitted). There are other subgoals in the higher levels, including stone pickaxe, iron, stone sword, fountain, iron pickaxe, iron sword, exorcism and killing zombie. Visiting some lower-level subgoals in certain orders can unlock and enable subgoals in higher levels.

The subgoal hierarchical structures are shown Figure 15, which are unknown to the agent initially. For example, when the agent first visits the stone and then tool workshop, the subgoal of making a stone pickaxe becomes feasible at the workshop, which can produce a stone pickaxe if selected. In order to make iron, coal and iron ore can be collected in any order and then fed into the iron workshop to produce iron. The prerequisites of both stone and iron swords contain wizard, since adding some magic to the raw materials can make the swords more effective. Asking the necromancer to process the stone sword can make it strong enough to kill the zombie. As shown in the highest level, the ultimate goal of this game is to use the sword to kill zombie or produce exorcism to drive away zombie. The LTL task in this domain is composed by all the subgoals from level 0 to level 3.

### G.2   AssemblyZone

This domain is modified from the Zone domain (Vaezipoor et al., 2021) where the subgoal hierarchy is added. It is a simulated robotic environment where the robot dynamics is implemented by the Safety Gym (Ray et al.,

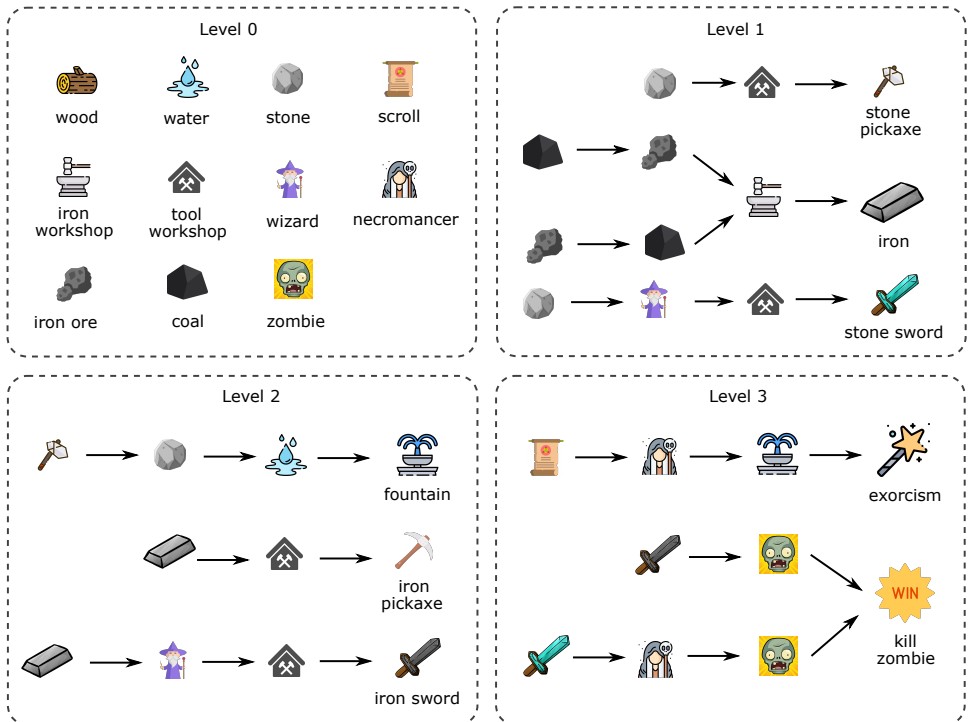

Figure 15: The subgoal hierarchy in Minecraft. Prerequisite sequences of subgoals are connected by arrows whose directions show the ordering constraints.

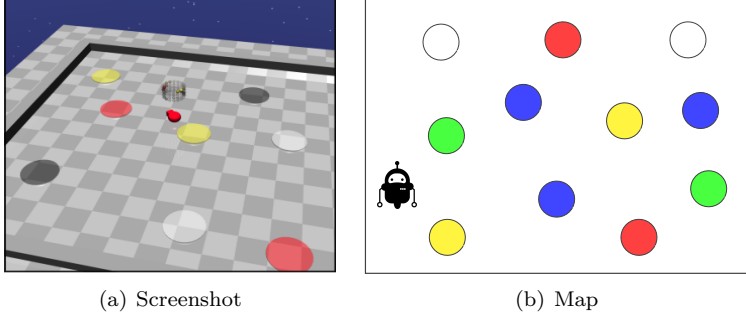

(a) Screenshot        (b) Map

Figure 16: Screenshot (left) and Map (right) of AssemblyZone.

2019). Specifically, the agent is the point robot with actions for steering and forward/backward acceleration. It observes lidar information towards the zones and other sensory data (e.g., accelerometer, velocimeter), where the observation is a real-valued vector with 92 dimensions. The zones and the robot are randomly positioned on the plane at the start of each episode. The agent is navigating in the map to visit zones in different colors in an order specified the LTL task, including red ("R"), blue ("B"), yellow ("Y"), green ("G") and white ("W"), as shown in Figure 16.

This domain is created to simulate the warehouse robot in practice. Zones in the same color contain same kind of parts. Whenever the agent visits a zone, it collects the corresponding part and gets a letter as a symbol. Collecting and assembling different parts in certain orders, which form sequences of symbols, can create new products. In addition, some products can be combined to produce more complex products. Therefore, parts in zones and products can form hierarchical structures of subgoals, as shown in Figure 17. The products numbered from 1 to 5 (short for "p1" to "p5") are categorized into two levels. The prerequisites of a product only contain products in lower levels. For example, after visiting the red, blue and green zones

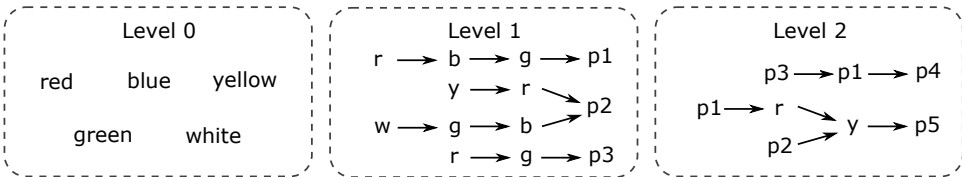

Figure 17: Hierarchical structures of subgoals in AssemblyZone. The subgoal prerequisites are connected by

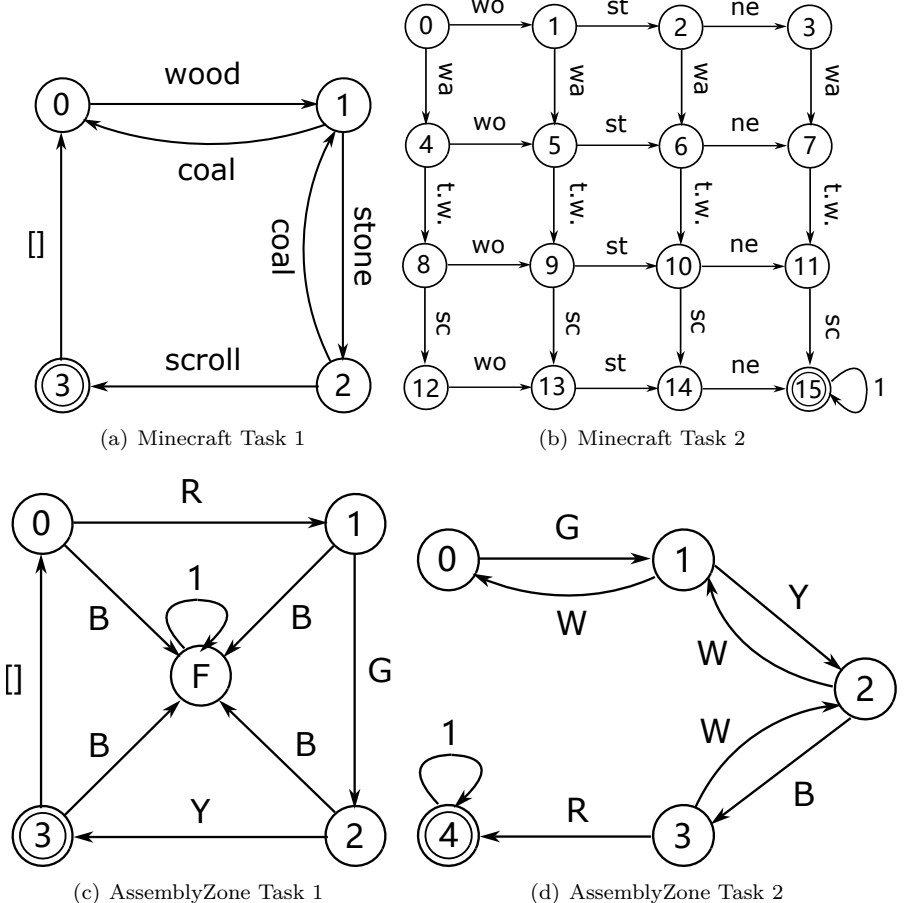

Figure 18: Task automaton compose by primitive subgoals. "wo", "st", "ne", "wa", "t.w.", and "sc" represent wood, stone, necromancer, water, tool workshop, and scroll, respectively. "R", "G", "B", "W" and "Y" are short for red, green, blue, white and yellow zones, respectively.

## H    Evaluation Tasks

The tasks composed by primitive subgoals are shown in Figure 18, where subgoals can be visited without satisfying any prerequisites. In the 2D-Minecraft environment, the task 1 for evaluation is to keep collecting wood, stone and scroll in a fixed order and collect collect a coal can go back to a previous automaton state. The task 2 is to first collect wood, second collect stone, then visit necromancer, and first collect

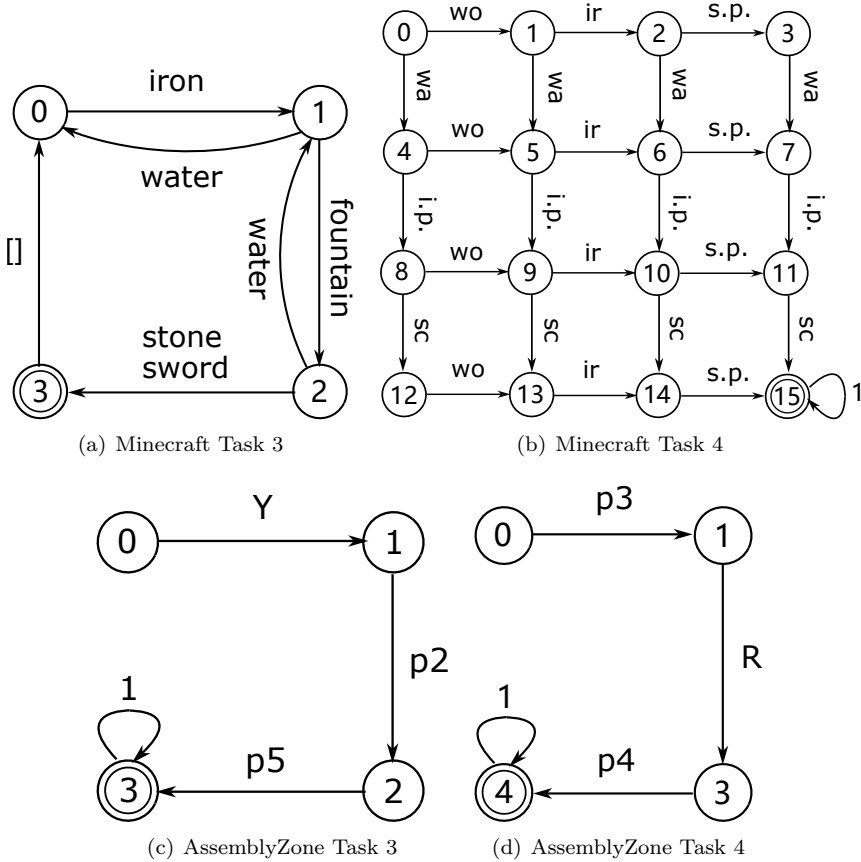

Figure 19: Task automaton compose by higher-level subgoals. In addition to Figure 18, "ir", "s.p." and "i.p." are short for iron, stone pickaxe and iron pickaxe, respectively.

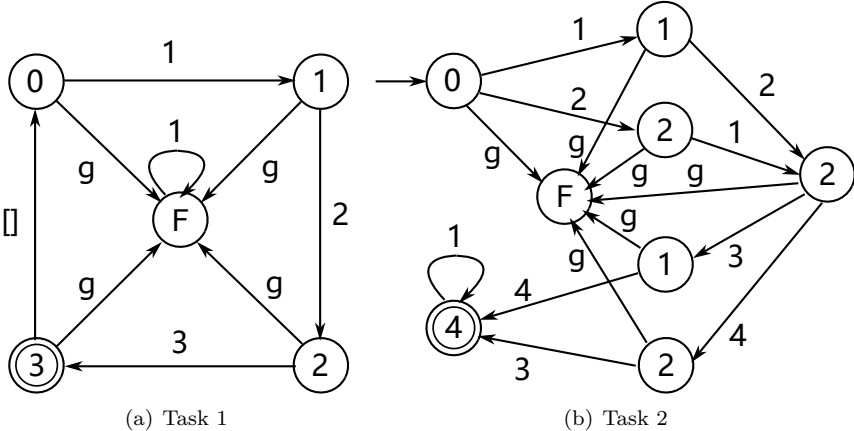

Figure 20: Task DFA in Button domain. Here numbers are indices of buttons and "g" represents the moving gremlin.

water, second visit tool workshop, then collect scroll, where these 6 items can be done in any orders as long as the ordering constraints are satisfied. Specifically, the task 2 of Minecraft can be represented as $\Diamond(\text{wo} \wedge \Diamond(\text{st} \wedge \Diamond\text{ne})) \wedge \Diamond(\text{wa} \wedge \Diamond(\text{t.w.} \wedge \Diamond\text{sc}))$ in the form of LTL formula. The task 1 in AssemblyZone environment is to keep visiting red, then green, then yellow zones in a fixed order without visiting blue zone at any time. It can be written as $\Box(R \wedge \Diamond(G \wedge \Diamond Y)) \wedge \Box(\neg B)$. The task 2 in AssemblyZone is to visit green, then yellow, then blue and finally red zone for only once, and visiting white zone can go back to a previous automaton state.

The tasks with subgoal hierarchy are shown in Figure 19, where visiting some subgoals needs to satisfy prerequisites first. The agent needs to form a product automaton to complete the task, which multiplies the task automaton and prerequisite automaton together, as an example shown in Figure 4. In Minecraft environment, task 3 and 4 are formed by replacing some subgoals in task 1 and 2 with higher-level subgoals. In AssemblyZone, task 3 and 4 are formed in similar ways.

The tasks in Button domain are shown in Figure 20.

## I Baselines

### I.1 Baselines for Hierarchy Discovery

In order to evaluate the performance on the exploration and discovery of subgoal hierarchy, we compare the stage 1 of HDCS with three baselines. Every method is evaluated in the same subgoal set $\mathcal{G}$. Subgoals in level 0, stored in the set $\mathcal{G}_0$, do not have any prerequisites to visit and are visible in the map, while visiting subgoals in higher levels has prerequisites in the form of subgoal sequences. The first baseline is used to evaluate the capability of exploration only, while other baselines are designed for both exploration and discovery of subgoal hierarchy.

- **Vanilla HRL (VHRL).** This is implemented as a two-level HRL agent. Specifically, in the low level, there is a goal-conditioned policy trained to achieve subgoals in $\mathcal{G}_0$ selected by the high-level part. The low-level action is the primitive environmental action. In the high level, an RNN-based policy is trained to learn sequences of subgoals in $\mathcal{G}_0$ to reach the final target of the environment (i.e., subgoals in the highest level), where the action space is $\mathcal{G}_0$ and the observation is the symbolic observation denoting the subgoal achieved by the agent.

- **HAC (Levy et al., 2019).** This method is a two-layer HRL similar as VHRL. The low-level part is same as that of VHRL. In contrast to VHRL, the high-level part is an goal-conditioned RNN-based policy trained to achieve a subgoal in $\mathcal{G} \setminus \mathcal{G}_0$ by using sequences of subgoals in $\mathcal{G}_0$. Besides, HAC discovers subgoal hierarchy with a randomness-driven exploration paradigm (Levy et al., 2019). We also use HER (Andrychowicz et al., 2017) to augment the training of the high-level part.

- **MEGA (Pitis et al., 2020).** This baseline is designed to use curriculum learning to improve HAC. MEGA train subgoals in $\mathcal{G} \setminus \mathcal{G}_0$ in the order of their training progress, where the subgoal in every episode is selected according to the principle of minimum density approximation. Other parts are same as HAC.

In every method evaluated in experiments, there is a pre-training phase which trains the low-level part to achieve subgoals in $\mathcal{G}_0$ in a fixed number of time steps. In HAC and MEGA, whenever well trained, the prerequisites of every subgoal can be obtained as the output sequences of the high-level policy conditioned on that subgoal, building the subgoal hierarchy.

### I.2 Baselines for Task Completion Evaluation

In every baseline, auxiliary reward is added to the original reward function of the environment, and the backbone is the PPO algorithm (Schulman et al., 2017) whose hyper-parameters are set same as the proposed framework. In Appendix 4.3, we compare critic shaping with these baselines in the Button domain, where the backbone is DDPG (Lillicrap et al., 2015).

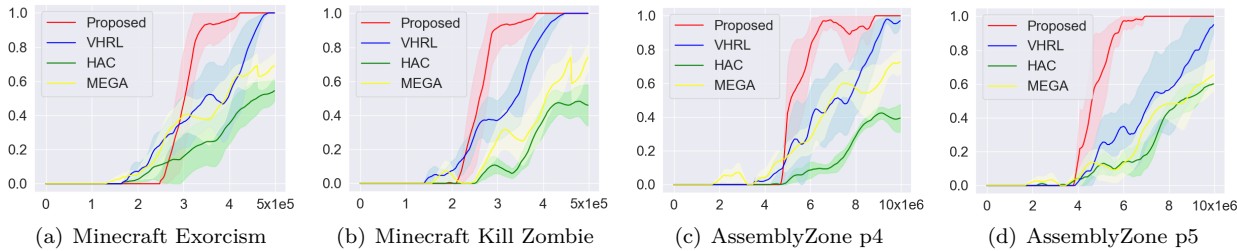

Figure 21: Performance comparison on the exploration capability of reaching a single final target of each domain. The success rate (y-axis) is about reaching one specific target denoted in the caption.

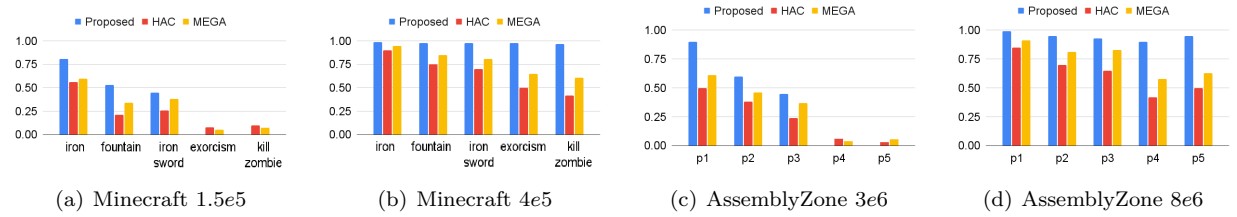

Figure 22: Performance comparison on reaching intermediate subgoals given fixed number of training samples. The number of training samples is denoted in the caption.

- **PBRS.** The first baseline is the PPO augmented with auxiliary rewards computed by using PBRS method (Ng et al., 1999; Jiang et al., 2021). The potential function is defined in terms of the automaton state, i.e., $\Phi(s)$. Specifically, $\Phi(s)$ is equal to the negative of distance from $s$ to its closest accepting state, multiplied by a scaling parameter $\kappa = 10$. An example with $\kappa = 1$ is shown in Figure 12(a).

- **Naive-RS.** Moreover, we adopt a naive baseline which rewards transitions over task automaton that reduce the distance to the accepting state, short as Naive-RS. For any state $u$ in task automaton $\mathcal{A}_\varphi$, the distance to its closest accepting state over $\mathcal{A}_\varphi$ is represented by the potential function $\Phi(u)$. For any transition $(\langle s, u \rangle, a, \langle s', u' \rangle)$, the auxiliary reward is 10 if $\Phi(u) > \Phi(u')$ is otherwise 0.

- **Adaptive-RS.** The third baseline we use is an adaptive variation of Naive-RS. It works in the same way as Naive-RS, but it adaptively adjusts auxiliary rewards of each stage by using the method introduced in Section 3.2.4, making every auxiliary reward adaptive to the difficulty of its corresponding stage.

## J   Subgoal Hierarchy Discovery

In this section, as a supplement to Section 4.1, we present more detailed results on the performance comparison about the discovery of subgoal hierarchy.

As shown in Figure 21, the proposed HRL-based method outperforms baselines in every highest-level subgoal of each domain, reaching every subgoal faster than baselines. Besides, given different number of training samples from the environment, we also evaluate the efficiency of the proposed and baseline methods on reaching some intermediate subgoals, as shown Figure 22. We can see that the proposed method still reaches intermediate subgoals faster than baselines. Note that baselines can reach highest-level subgoals with some probability in the early learning stage. This is because these baselines select the target subgoals to train the high-level agent with some randomness, ignoring the hierarchical nature of the subgoal space. However, this ignorance makes these baselines be outperformed by the proposed method finally.

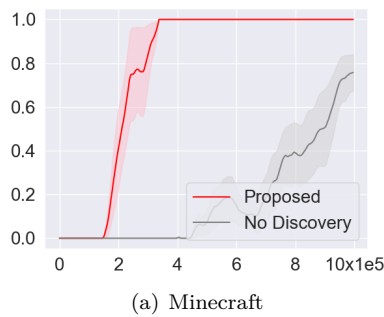 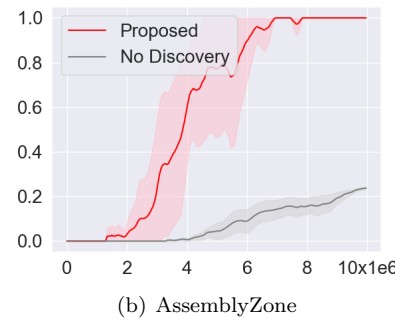

(a) Minecraft          (b) AssemblyZone

Figure 23: Ablation study on subgoal hierarchy discovery. The tasks for evaluation are both task 4 in Minecraft and AssemblyZone domains.

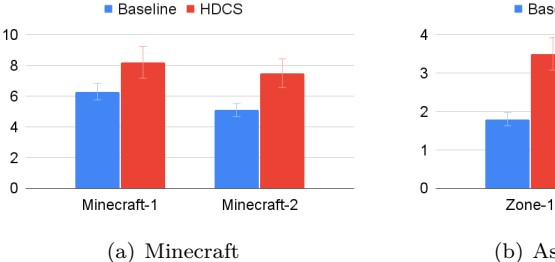

(a) Minecraft          (b) AssemblyZone

Figure 24: Optimality comparison. y-axis is the return (discounted accumulated rewards).

## K  Ablation Study

In this section, we will conduct some ablation studies on the HDCS framework. We first investigate the performance of task completion without subgoal hierarchy discovery, and then compare different choices of auxiliary rewards which may violate the condition equation 2, showing the importance of various components of the algorithm design.

### K.1  Effect of Hierarchy Discovery

In order to study the effect of hierarchy discovery, we design a baseline "No Discovery" which directly uses the second part of HDCS framework, i.e., critic shaping and PPO, to solve the task 4 of Minecraft and AssemblyZone. These two tasks contain subgoals which have prerequisites to satisfy. The performance comparison is shown in Figure 23. We can see that the baseline almost cannot solve the given task, showing the significance of subgoal hierarchy discovery.

### K.2  Ablation Study for Optimality

A simple idea is to use the low-level policies trained in phase 1 to directly solve the given DFA task in phase 2, without using critic shaping to train any new policies. However, this method can have the sub-optimality issue in the found solutions. We term this method as "baseline", and compare its optimality with the proposed HDCS framework. The comparison is shown in Figure 24.

In Figure 24, we first compare the baseline and HDCS in task 1 and 2 in 2D-Minecraft domain, short as Minecraft-1 and Minecraft-2. We also compare the optimality in task 1 and 2 in AssemblyZone domain, short as Zone-1 and Zone-2. The DFA of these tasks are presented in H. The final reward $R_F$, is given at the last visit to an accepting state in task DFA, where $R_F = 15$ in Minecraft and 100 in AssemblyZone. Auxiliary rewards are not counted when computing the return. The higher the return is, the less time steps are used in task completion.

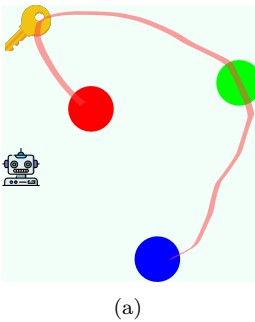 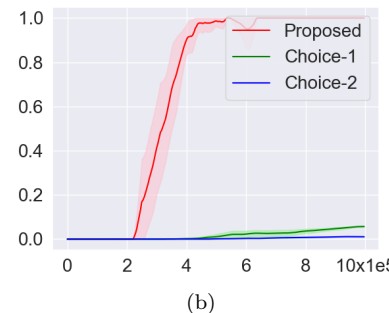

(a)                                        (b)

Figure 25: Ablation study on different auxiliary rewards in AssemblyZone domain. (a): The red path shows the optimal solution of the task. (b): performance comparison for abalation study.

In baseline, no policies are trained in the phase 2. In HDCS, the return is evaluated when the agent is trained with $5e5$ and $10e6$ environmental samples in Minecraft and AssemblyZone domains, respectively. We can see that the HDCS outperform the baseline in every evaluated task. This is because every symbol has multiple copies in both domains and the policies in HDCS are trained from scratch to select the optimal symbol to visit.

### K.3 Effect of Auxiliary Reward Condition

In this section, we study the importance of the auxiliary reward condition equation 2. We design a new task in the AssemblyZone domain by introducing some causal relationship. As shown in Figure 25(a), the task is to first visit the red, then green, and finally blue zones, where each zone has fixed position in every episode. There is a yellow key (relized by a yellow zone) fixed at the upper left corner of the map. The sizes of map and zones are chosen smaller than previous experiments in AssemblyZone domain.

However, we assume that the agent can only visit blue zone with a yellow key which can be picked up by visiting the yellow zone. The agent does not know this causal relationship initially, and it has to learn this by interacting with the environment. The horizon of every stage is 300 and discount factor is 0.998. In this case, we apply critic shaping and PPO , which is the stage 2 of HDCS, to solve the given task and evaluate different choices of auxiliary rewards $R_A$. The proposed framework chooses $R_A = [10, 20, 40]$ for three stages in task DFA. Additionally, we have $[10, 10, 10]$ as choice-1, and $[10, 30, 10]$ as choice-2.

The performance comparison is shown in Figure 25(b). We can see that these two choices perform poorly and cannot solve the task at all. The reason is that the auxiliary rewards of stage 2 and 3 do not satisfy condition equation 2 and prevent the reward information of stage 3 (i.e., reaching the blue zone) from back-propagating to stage 2. This can make the agent only focuses on reaching green zone in stage 2 and ignore picking up the key, so that the blue zone can never be reached in stage 3. If the reward cannot back-propagate across every stage smoothly, the optimal solution cannot be learned.

### K.4 Effect of Max Operator

In addition, we further conduct ablation studies on the max operators in critic shaping. The baseline method replaces the max operator in equation 3 and equation 4. The environment is the AssemblyZone domain and the tasks are task 1 and 2 without subgoal hierarchy (introduced in Appendix H). We choose $R_A = [10, 20, 40, \ldots]$. The performance comparison is shown in Figure 26.

We can see that if the proposed method replaces the max operator with the addition used in conventional reward shaping methods, the performance degrades. The reason is that, although auxiliary rewards $R_A$ are large enough to satisfy equation 2, the values produced by the critic function are usually small in the early learning stage. Using the max operator can prevent the learning in one stage from being influenced by the values of next stage. Hence, it stabilizes the targets of the learning of critic function and improves the

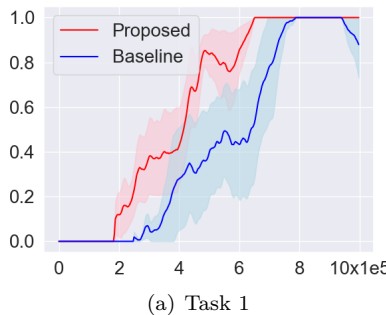
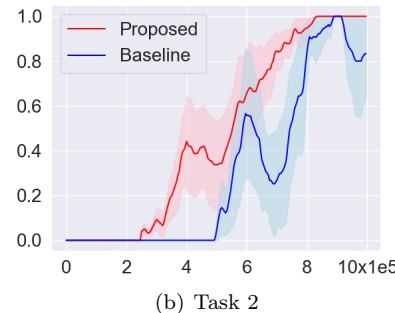

(a) Task 1

(b) Task 2

Figure 26: Ablation study on the max operator of critic shaping in the AssemblyZone domain.

learning efficiency of both critic and policy. This is visualized and demonstrated by the blue curve in Figure 13 at Section F.

