# OpenReview forum: "HDCS: Hierarchy Discovery and Critic Shaping for Reinforcement Learning with Automaton Specification"
_TMLR — Accepted by TMLR_

### Review · Reviewer_LooW · 2025-05-16

**Summary Of Contributions:**

This paper studied the problem of using deterministic finite-state automaton (DFA) to define complicate tasks which are not easy to defined by a single scalar reward. The authors identified two problems within the current framework: (1) There can be hidden temporal structure about the fulfillment of goals and (2) the sparse reward problem when the DFA is complicated. For the first problem, the authors propose an exploration method to build the hierarchical structure of the hidden DFA iteratively; While for the second problem, the authors propose a critic-shaping method based on heuristic. Experiments on three toy examples demonstrate the proposed method is better than some baselines.

**Audience:**

Yes

**Broader Impact Concerns:**

None.

**Claims And Evidence:**

No

**Requested Changes:**

See the Weakness section.

**Strengths And Weaknesses:**

## Strength
- The problem studied seems to be relevant for complicated problems in RL.
- The general idea behind the method sounds reasonable.

## Weakness

The biggest issue for this paper is that the presentation is very confusing, which makes it hard to evaluate the value of the proposed method. Here are some questions of mine:
- For the first phase, the exploration, the authors seem to assume the availability of a low-level policy $\pi^l(a|g)$ which can reach all the goals already. How is this policy obtained in the experiments? More importantly, how would such a policy be acquired in real-world applications?
- After the exploration phase, we obtain a hierarchical policy that can reach any goal given the correct prerequisites. Given a task DFA, wouldn't executing the appropriate sub-policies in sequence suffice to solve the task? It's unclear what is being learned in the second phase.
- Figure 5 is very confusing. Can you explain what does the different colour means for the arrows?
- I am not sure if I understand the product DFA correctly. There are drawings for task DFA and prerequisite DFA, but not for product DFA. Can I understand it as a flattened and merged version of the task DFA and prerequisite DFA? For instance, for the example in Figure 11, should I understand the product DFA as a sequence of a → b → c → f → e → c that the agent need to visit in the correct order?
- The difference between the proposed method and the relevant baselines is not clear to me. For the first phase, from Appendix G.1, it seems like the only difference is that the proposed method can build arbitrary layers of hierarchy while all the baselines are constrained to build only 2 layers.
- Why is VHRL not in Figure 6 (c, d) and Figure 7?
- Why is the number of environment interactions different between figures? For Minecraft, Figure 6 (a) shows up to $5e5$, Figure 6 (c) shows the point at $3e5$ while Figure 7 (a) shows up to $1e6$; For AssemblyZone, Figure 6 (b) and Figure 7 (b) show up to $1e7$ while Figure 6 (d) shows at the point of $6e6$. How are these numbers selected?

Minor issues:

- Citation Format issue: I would like to remind the author to take care of the difference between \citep and \citet, the majority of the citations in this paper should in \citep format but they are currently all in \citet format.
- I think the SC2 example in Figure 1 is a bit misleading. First, the hierarchy there is very subjective. For example, to scout the enemy defence one can do a radar-scan instead sending a small group of units. Minecraft can be a better example since the hierarchy is objective. For example, one cannot build a stone pickaxe without stone, which is a hard constraint of the environment. Second, it feels like an overstatement since the algorithm in this paper feel far away for solving SC2 and you are not conducting experiments on that task.

---

> ### Author Response · Authors · 2025-06-10
> **Response to reviewer LooW**
>
> Thanks for your review and time on our work. My response to your questions are as below.
>
> 1. For the first phase, the exploration, the authors ......
>
> These are low-level policies to reach subgoals in the level 0. As we claimed, they are pre-trained before uncovering any higher-level subgoals. This pre-training stage explains why there is a long and flat line at 0 success rate in every plot.
>
>
> 2. After the exploration phase, we obtain ......
>
> The second phase is necessary because of some optimality issue and changes of environment in the test. First, executing sub-policies to reach subgoals one-by-one may obtain a sub-optimal or even infeasible solution of the DFA task, especially when there are multiple copies of every subgoal in the environment. The paper [1] shows a good example in Figure 1. If directly applying sub-policies of subgoals independently to solve the task, the agent may choose to go to the wrong copy of the blue subgoal and cannot solve the task at all. Therefore, it is important to treat the DFA task as a whole and train the policy of solving the DFA task from the scratch in the second phase.
>
> In addition, if directly applying sub-policies of subgoals, the agent cannot adapt to changes of the environment when the DFA task is given, since these sub-policies are all trained in the old environment.
>
> [1] Vaezipoor, Pashootan, et al. "Ltl2action: Generalizing ltl instructions for multi-task rl." International Conference on Machine Learning. PMLR, 2021.
>
>
> 3. Figure 5 is very confusing. Can you explain what does the different colour means for the arrows?
>
> This figure exemplifies the concepts of level, stage, and round. We clearly demonstrate these concepts on page 8 and I can repeat them here again. The nodes are automaton states and edges are state transitions of DFA.
>
> The level of a state is its minimum number of hops to an accepting state, where green nodes are in level 2 and pink nodes are in level 1. The stage refers to the lowest level the agent has reached up to now, e.g., Red line: stage 2 and Blue line: stage. The round is the interval between last and next visit to an accepting state, e.g., green line: round 2.
>
>
> 4. I am not sure if I understand the product DFA correctly. There are ......
>
> Yes, your understanding is correct.
>
>
> 5. The difference between the proposed method and the ......
>
> The difference is that all the previous works/baselines did not consider the hierarchy of subgoals and treated every unknown subgoal equally. However, our method specifically builds the subgoal hierarchy level-by-level and discover hierarchy and prerequisites of subgoals more efficiently. As far as we know, no previous methods addressed the problem of building subgoal hierarchy specifically.
>
>
> 6. Why is VHRL not in Figure 6 (c, d) and Figure 7?
>
> This is because VHRL did not learn to reach every subgoal specifically and could not learn prerequisites of subgoals. In VHRL, the only target is to reach a subgoal in the highest level, which is designed to evaluate the exploration capability only. In Figure 6 (c, d) and Figure 7, the metric is the success rate of reaching an intermediate subgoal, which is beyond the capability of VHRL.
>
>
> 7. Answer: We clearly explain the meaning of these numbers in Section 4.1 and Appendix. These numbers are numbers of training samples (i.e., experience tuples) drawn from the environment. Their differences can be explained by the fact that solving problem in a more difficult environment needs more samples drawn from that environment.
>
> The scales of x-axis in Figure 7 (a) and (b) are selected according to when the proposed method can reach its best and steady performance. Specifically, the final target in Minecraft (AssemblyZone) can be steadily reached by the proposed method with $5e5$ ($10e6$) samples from the environment. This difference is from the differences of the environments. The scale of x-axis in AssemblyZone is always larger than that in Minecraft, since it is more difficult to solve.
>
> In Figure 6 (c) and (d), the performance is evaluated when there are $3e5$ and $6e6$ samples drawn from the environment, respectively. In these figures, we just want to compare the performance of reaching intermediate subgoals. The performance difference is more significant in the middle of the training process. We select $3e5$ and $6e6$, since they are relatively in the middle of the whole x-axis.
>
> In Figure 7 (a) and (b), we compare the accuracies of learned subgoal prerequisites in Minecraft and AssemblyZone. The x-axis of Figure 7 (b) can reach $10M$ while that of Figure 7 (b) can only reach $1M$. This is because the agent in AssemblyZone is more difficult to control. The agent needs much more samples to learn subgoal prerequisites in AssemblyZone than Minecraft.

---

> > ### Comment · Reviewer_LooW · 2025-06-16
> > **Reply to rebuttal**
> >
> > I would like to thank the authors for their response. However, there are still a few points that remain unclear to me:
> >
> > - **Pretraining of Low-Level Policies (Phase 1):** The only mention of this pretraining appears in Appendix G.1, but there are no details provided about how it is implemented. Please include sufficient information about the pretraining procedure in the main text or appendix.
> >
> > - **Phase 2 and Environment Changes:** I agree that directly using the pretrained policy may be suboptimal, and I think it can be beneficial to include it as a baseline for Phase 2. However, I am unclear about your argument regarding potential changes in the environment. As I understand it, the environment should remain the same to ensure that the learned prerequisites are still valid in Phase 2. Moreover, I did not find any experiments in the paper that involve a change of environment. Could you please clarify what you mean by "changes of the environment"?
> >
> > - **Figure 5:** I believe there was a misunderstanding regarding my previous question. I was referring to the **color of the arrows** in the figure—some are green while others are black. Could you please explain the significance of this color difference?
> >
> > - **Figures 6 and 7:** Thank you for your explanation regarding the numbers in Figures 6 and 7. Based on your clarification, I would recommend extending Figure 6(a) to 1M steps to align with Figure 7(a), which would help reduce potential confusion.
> >
> >
> > In addition to the above points, I kindly ask the authors to incorporate their responses into a revised version of the paper, clearly marked with a different color, and upload it to the OpenReview system. I will only consider revisions submitted through this official channel, as this ensures compliance with the double-blind review policy.

---

### Review · Reviewer_ZGCC · 2025-05-18

**Summary Of Contributions:**

The paper presents **HDCS (Hierarchy Discovery and Critic Shaping)**, a reinforcement learning framework that automatically discovers sub-goal hierarchies and improves learning efficiency in tasks specified by deterministic finite-state automata (DFA). HDCS first learns prerequisite DFA structures through hierarchical policy interactions, effectively capturing intermediate sub-goals without manual specification. Next, the paper introduces a critic-shaping approach that mitigates sparse reward issues by providing more stable and dense training signals without altering optimal policies. Empirical evaluations conducted across three simulation domains—2D-Minecraft, AssemblyZone, and Safety-Gymnasium—demonstrate HDCS’s superiority over existing hierarchical RL and reward-shaping baselines in terms of both accuracy of hierarchy discovery and overall sample efficiency. Extensive ablations reinforce the significance of each component of the proposed method.

**Audience:**

Yes

**Broader Impact Concerns:**

The work aims to improve interpretability and effectiveness in specifying complex, temporally-extended tasks, potentially benefiting safety-critical and robotics domains. Nevertheless, incorrect hierarchy discovery could inadvertently introduce unsafe constraints. The authors should explicitly discuss strategies to mitigate such risks, including human validation or automated verification processes. Otherwise, no significant ethical or societal issues are apparent.

**Claims And Evidence:**

Yes

**Requested Changes:**

1. Provide a formal theoretical analysis or proof clarifying under what conditions the critic-shaping method preserves optimal policies.
2. Conduct a hyperparameter robustness analysis, focusing on parameters like stage horizon and auxiliary reward scale, to assess the sensitivity of HDCS's performance.
3. Evaluate HDCS on more public benchmarks (e.g., MiniGrid-LTL, CraftWorld or ZoneENV...) to demonstrate broader applicability.

**Strengths And Weaknesses:**

The paper offers a clear and innovative approach by coupling automated hierarchy discovery with critic shaping, significantly improving sample efficiency on sparse-reward tasks. Its major strength is the careful empirical validation: multiple domains, strong baselines, detailed ablations, and openly available code support reproducibility and thoroughness. The introduced critic-shaping technique is conceptually simple yet notably effective, enhancing practicality.

However, several concerns exist. The theoretical justification for why the critic-shaping technique preserves optimality remains informal, lacking explicit proofs or rigorous conditions. Evaluation tasks are limited to custom-made simulations, leaving open questions about the approach’s generalization to widely-used benchmarks or real-world applications. Additionally, the scalability of the method with respect to computational resources and state-space complexity is not clearly addressed. Important recent baselines relevant to DFA-based reinforcement learning are also absent from comparisons.

---

### Review · Reviewer_1PEB · 2025-05-26

**Summary Of Contributions:**

The paper tackles two long-standing obstacles in automaton-specified reinforcement learning (RL), including hidden sub-goal hierarchies and extreme reward sparsity.
The authors propose HDCS, a two-phase framework with sub-goal hierarchy discovery and critic shaping. Experiments on three domains (2D-Minecraft, AssemblyZone, Safety-Gym Button) show 2–3 × faster convergence versus PBRS, Naive-RS, Adaptive-RS and several HRL baselines.

**Audience:**

Yes

**Claims And Evidence:**

Yes

**Requested Changes:**

- provide O(#states, #sub-goals) bounds for Phase 1 exploration and Phase 2 convergence.
- please add a table to map prior methods with the addressed challenges, and highlight what HDCS uniquely contributes.
- It will be helpful to add experiments to show the real scaling, e.g. MiniGrid-SkillWorld or StarCraftII mission

**Strengths And Weaknesses:**

Strength:
- As far as I know, unknown hierarchies & sparse rewards are rarely handled together in existing
- The method, bypassing slow reward-to-critic propagation and yielding stable learning targets, seems interesting
- Composite ablation studies look good, including (i) omitting hierarchy discovery, (ii) illegal auxiliary-reward schedules, and (iii) removing the max, each of which degrades performance
- Code link is provided; hyper-parameters and task DFAs are spelled out in appendices.

Weakness:
- The main concern is that the experiment platform is oversimple, Minecraft and AssemblyZone are handcrafted, and the Button tasks have $\leq$ 4 buttons. It is unclear how does HCDS scales to larger DFAs ($\gg$ 20 states) or real robotics.
- The auxiliary-reward schedule $R_A(i,k)$ is hand-tuned per domain (Appendix D). How robust is HDCS if these values are mis-specified, especially in unseen environments?
- Phase 1 pre-training seems sample-hungry (millions of steps, Fig. 6). A wall-clock or GPU cost comparison is missing.
- The critic-shaping idea resembles “max-reward RL” (Veviurko et al., 2024) and the hierarchy discovery phase borrows heavily from HAC/MEGA but with a level-wise schedule.

---

### Decision · Action_Editor_xpm8 · 2025-07-02

**Recommendation:** Accept as is

**Additional Comments:**

All three reviewers recommend acceptance (one Accept, two Leaning Accept), recognizing the paper's technical merit and relevance. The work tackles automatic discovery of subgoal hierarchies in DFA tasks; and provides a novel critic shaping approach that preserves optimal policies.
Initial concerns about theoretical justification, experimental platforms, and presentation clarity were adequately addressed through author responses and revisions. The authors provided convergence proofs, clarified experimental setups, improved figures, and added comprehensive related work sections. While some limitations remain (simple experimental environments, limited scalability evidence), reviewers agreed these don't diminish the core contributions.
The thorough ablation studies, comparison with relevant baselines, and open code availability strengthen the work's reproducibility and impact potential.

**Audience:**

Yes

**Audience Explanation:**

Yes, individuals in TMLR's audience would be interested in this work. The paper addresses relevant challenges in reinforcement learning with temporal logic specifications, particularly for robotics and safety-critical applications. The combination of hierarchy discovery with critic shaping for DFA-specified tasks represents a timely contribution to the RL community, especially given the growing interest in structured task specifications.

**Claims And Evidence:**

Yes

**Claims Explanation:**

Yes, the claims are supported by convincing evidence. The authors present HDCS, which addresses two key challenges in DFA-based RL: discovering hidden subgoal hierarchies and mitigating sparse rewards through critic shaping. The theoretical analysis includes a convergence proof for the critic shaping method, and empirical results across three domains (2D-Minecraft, AssemblyZone, Safety-Gym Button) demonstrate improvement compared to baselines. The ablation studies and open-source code further support reproducibility.